# Lysosomal SLC46A3 modulates hepatic cytosolic copper homeostasis

Jung-Hwan Kim [1,2✉], Tsutomu Matsubara [2,7], Jaekwon Lee[3], Cristina Fenollar-Ferrer[4], Kyungreem Han[5], Donghwan Kim[2], Shang Jia[6], Christopher J. Chang [6], Heejung Yang [2,8], Tomokazu Nagano[2,9], Kristopher W. Krausz[2], Sun-Hee Yim[2,10] & Frank J. Gonzalez [2✉]

The environmental contaminant 2,3,7,8-tetrachlorodibenzo-p-dioxin (TCDD) causes hepatic toxicity associated with prominent lipid accumulation in humans. Here, the authors report that the lysosomal copper transporter SLC46A3 is induced by TCDD and underlies the hepatic lipid accumulation in mice, potentially via effects on mitochondrial function. SLC46A3 was localized to the lysosome where it modulated intracellular copper levels. Forced expression of hepatic SLC46A3 resulted in decreased mitochondrial membrane potential and abnormal mitochondria morphology consistent with lower copper levels. SLC46A3 expression increased hepatic lipid accumulation similar to the known effects of TCDD exposure in mice and humans. The TCDD-induced hepatic triglyceride accumulation was significantly decreased in $Slc46a3^{-/-}$ mice and was more pronounced when these mice were fed a high-fat diet, as compared to wild-type mice. These data are consistent with a model where lysosomal SLC46A3 induction by TCDD leads to cytosolic copper deficiency resulting in mitochondrial dysfunction leading to lower lipid catabolism, thus linking copper status to mitochondrial function, lipid metabolism and TCDD-induced liver toxicity.

[1] Department of Pharmacology, School of Medicine, Institute of Health Sciences, Department of Convergence Medical Science, Gyeongsang National University, Jinju 52727, Republic of Korea. [2] Laboratory of Metabolism, Center for Cancer Research, National Cancer Institute, National Institutes of Health, Bethesda, MD 20892, USA. [3] Department of Biochemistry, University of Nebraska-Lincoln, Lincoln, NE 68588, USA. [4] Laboratory of Molecular & Cellular Neurobiology, National Institute of Mental Health, National Institutes of Health, Bethesda, MD 20892, USA. [5] Laboratory of Computational Biology, National Heart, Lung and Blood Institute, National Institutes of Health, Bethesda, MD 20892, USA. [6] Departments of Chemistry and Molecular and Cell Biology, University of California, Berkeley, CA 94720, USA. [7] Present address: Department of Anatomy and Regenerative Biology, Osaka City University Graduate School of Medicine, Osaka City University, Osaka, Japan. [8] Present address: College of Pharmacy, Kangwon National University, Chuncheon, Republic of Korea. [9] Present address: Sumitomo Dainippon Pharma Co. Ltd., Osaka, Japan. [10] Present address: Department of Environmental Toxicology, Texas Tech University, Lubbock, TX 41163, USA. ✉email: junghwan.kim@gnu.ac.kr; gonzalef@mail.nih.gov

Nonalcoholic fatty liver disease (NAFLD) is one of the most common chronic liver diseases in developed countries[1]. Fatty liver can be caused by diet, drugs, viruses, genetic factors, hormones, or environmental pollutants[2]. Hepatic lipid accumulation can lead to nonalcoholic steatohepatitis, cirrhosis, liver failure, and hepatocellular carcinoma. Abnormal hepatic lipids can also be caused by chemically-induced toxicity by compounds like dioxins that are constituents of environmental pollution that threaten the ecosystem and exist as complex isomers of aromatic halogen compounds[3]. The most poisonous toxicant among the dioxins is 2,3,7,8-tetrachlorodibenzo-p-dioxin (TCDD), a major contaminant in Agent Orange, an herbicide used in the Vietnam War between 1962 and 1971 to destroy forests for military purposes. TCDD is also produced during incomplete combustion of organic materials. Accidental exposure to dioxins induces tissue-specific toxicity with decreased immune function, increased hepatic drug metabolizing-enzyme induction, teratogenesis, thymic degeneration, cirrhosis, endocrine disruption, infertility, liver toxicity associated with increased lipid accumulation, and cancer[4,5].

TCDD toxicity is due to its ability to activate the aryl hydrocarbon receptor (AhR), a ligand-activated transcription factor that is activated by xenobiotics including dioxins and polycyclic aromatic hydrocarbons[6]. The molecular mechanisms of many specific toxicities elicited by TCDD exposure are still not fully understood. In experimental animal models, these effects are due to activation of the AhR. The AhR nuclear translocator (ARNT) and ligand-bound AhR form a heterodimer that binds to elements in the promoter regions of target genes and activates transcription[7,8].

The function of solute carrier 46a3 (SLC56A3) is not known. Slc46a3 mRNA, similar to the AhR-regulated Cyp1a2 mRNA, is constitutively expressed in livers of wild-type mice and upon treatment with TCDD, was induced with similar kinetics to Cyp1a2 mRNA; induction of both mRNAs was not found in Ahr-null and Arnt liver-specific conditional-null mice. In view of the effects of TCDD on hepatic steatosis, it was of interest to determine whether SLC46A3 expression and its induction by AhR, affects hepatic lipid levels. While phylogenetically SLC46A3 is a solute carrier, its function and substrate(s) are unknown. Thus, the function of SLC46A3 and it role in liver lipid homeostasis was studied using Slc46a3−/− mice, cultured primary hepatocytes, and forced expression of recombinant SLC46A3 in cultured cells. Induction of lysosomal SLC46A3 by TCDD induces intracellular copper deficiency, which results in mitochondrial dysfunction resulting in lower lipid catabolism and hepatic lipid accumulation.

## Results

**Lysosomal SLC46A3 is regulated by AhR.** Slc46a3 mRNA was induced by TCDD in the mouse liver while no induction was found in kidney or different regions of the small intestine and the colon, where constitutive Slc46a3 mRNA was detected, indicating liver-specific induction by the AhR (Fig. 1a). This is a similar pattern of induction found with another AhR target gene Cyp1a2, which shows liver-specific induction. Both Slc46a3 and Cyp1a2 mRNAs were induced in an AhR-dependent manner since no induction was found in Ahr−/− and ArntΔLiv mice (Fig. 1a, b). As a negative control for TCDD, the non-dioxin-like polychlorinated biphenyl-153 (PCB-153) was analyzed by extraction of Gene Expression Omnibus (GEO) data sets (accession: GSE55084), and no induction of Slc46a3 mRNA was noted (Fig. 1c)[9].

Putative DRE sites were found in a published ChIP-seq data set (GEO, accession: GSE97634) (Fig. 1d, e). A potential remote DRE site −17,135 bp of upstream of Slc46a3 was examined for trans-activation

by AhR and TCDD using luciferase reporter gene assays, revealing that 3×DRE-luciferase activity was significantly increased by 10 nM TCDD in Hepa1c1c7 cells and was abolished by mutation of the DRE site (Fig. 1f) and CH-223191, an AhR antagonist (Fig. 1g). Likewise, CH-223191 suppressed TCDD-induced Slc46a3 mRNA levels (Fig. 1h). These results suggest that AhR may be functioning through a DRE located in a remote long-range enhancer controlling AhR activation of the Slc46a3 gene.

**Time and dose-dependent expression of Slc46a3 mRNA induction by TCDD.** In mouse liver, Slc46a3 mRNA was developmentally gradually increased up to 8 weeks of age (Fig. 1i). In addition, Slc46a3 mRNA was elevated seven fold by 10 µg/kg TCDD in a time-dependent manner over 24 h (Fig. 1j). No significant loss of Slc46a3 mRNA levels was noted after 3 days and expression persisted for 7 days (Fig. 1k). This might be due to the high affinity of TCDD to AhR and its persistent activation, or due to a very stable Slc46a3 mRNA that, once induced, does not rapidly decay.

**Generation and phenotype of Slc46a3−/− mice.** To explore a role for SLC46A3 in the liver, Slc46a3−/− mice were generated (Fig. 2a–c). The mice were born at the expected frequency, showed no developmental defects, and no gross abnormal phenotype, as compared to matched wild-type controls. A striking color change was noted in the livers of Slc46a3−/− mice compared to wild-type mice. Notably, the livers of Slc46a3−/− mice were darker, and, after perfusion with saline, showed a more grayish color (Fig. 1d). However, H&E staining revealed no difference in both groups in the classic hepatic inflammation features in the livers of mice administered TCDD (Fig. 2e).

**Localization of SLC46A3.** SLC46A3 protein was localized in the lysosomal membrane (Fig. 2f) as revealed by lysosomes incorporated with eGFP-SLC46A3 that co-isolated with LAMP1 (Fig. 2g). The eGFP-SLC46A3 aggregated during denaturing by boiling as revealed by western blot analysis which indicates the marked hydrophobicity of SLC46A3, as expected from this class of membrane-bound proteins (Fig. 2h).

**Copper is a possible substrate of lysosomal SLC46A3.** Based on the color change in the liver of Slc46a3−/− mice, hepatic metal contents were measured in WT and Slc46a3−/− mice (Supplementary Fig. 1a). Among the physiologically relevant metal ions, copper levels were notably increased by around 30% in perfused livers of Slc46a3−/− mice, with no differences in iron contents (Fig. 3a and Supplementary Fig. 1b). This observation might be due to lysosomal copper accumulation. However, liver copper-related mRNA levels were not changed after 7 days of 10 µg/kg TCDD (Fig. 4f). For lysosomal copper analysis, hepatic lysosomes were isolated using magnetic beads after hydrodynamic injection with eGFP-LAMP1 and Dsred or Dsred-SLC46A3. The lysosomal fraction was isolated only with the N-terminal eGFP tagged LAMP1 (Supplementary Fig. 2). Copper levels were significantly increased by Dsred-SLC46A3 from the isolated lysosome (Fig. 3b). Furthermore, the 13,000 g pellet from the lysate of Hepa1c1c7 cells had elevated FLAG-SLC46A3 protein levels (Fig. 3c), and copper levels were significantly increased in the presence of copper sulfate (Fig. 3d). Based on copper analysis using liver and cell lines, SLC46A3 may transport copper into the lysosome. Thus, a quenching effect was tested in lysosomes using Phen Green FL dye in the presence of recombinant SLC46A3 expression in Hepa1c1c7 cells. A quenching effect was found in lysosomes embedded with mCherry-SLC46A3 (Fig. 3e) as revealed by decreased levels of Phen Green fluorescence in the

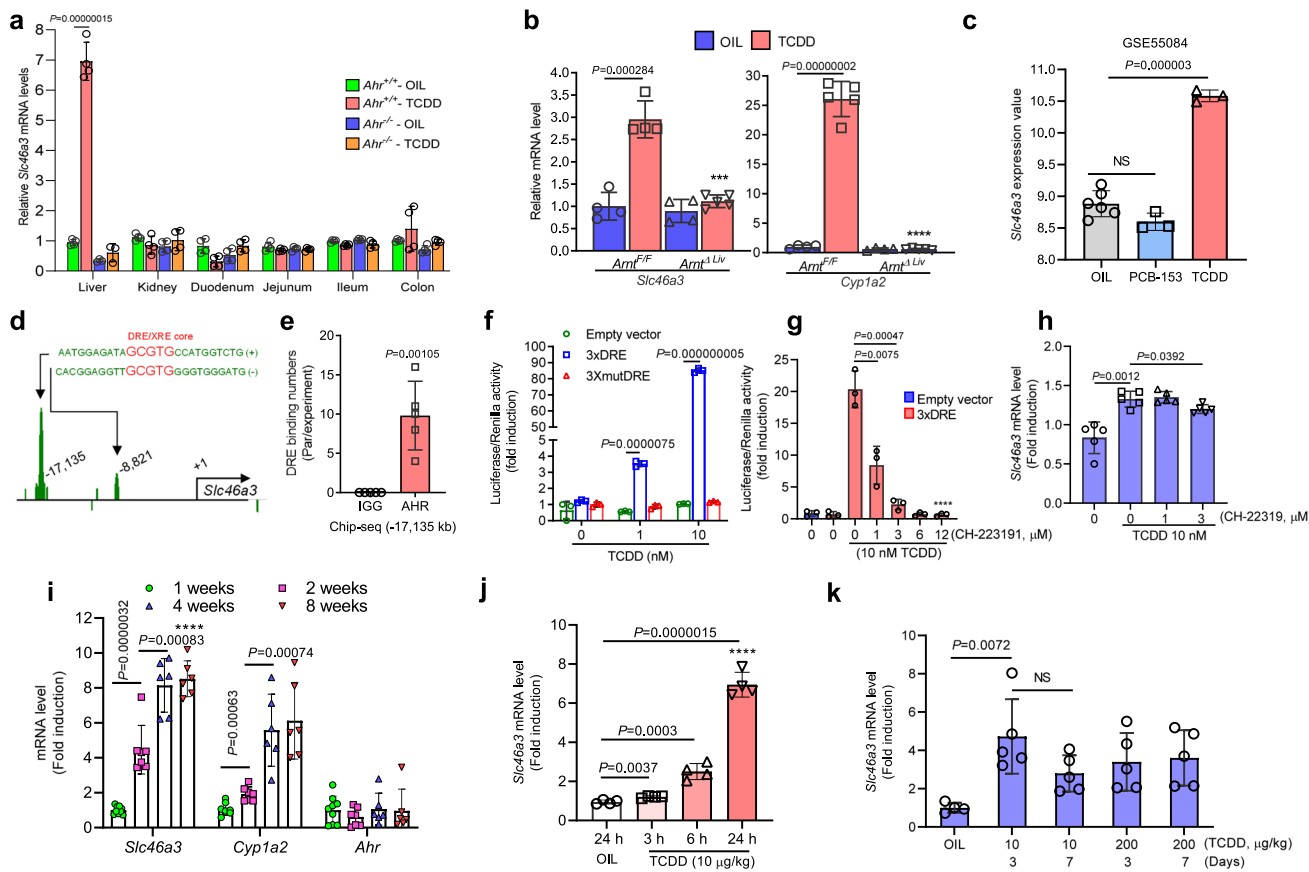

**Fig. 1 Regulation of *Slc46a3*. a** *Slc46a3* mRNA expression in different mouse tissues after oil ($n = 3$–4 per group) or TCDD (10 μg/kg for 24 h)($n = 3$–4 per group) administration. **b** Hepatic *Slc46a3* mRNA expression after oil ($n = 4$ per group) or TCDD (10 μg/kg for 24 h)($n = 5$ group) administration in $Arnt^{-/-}$ mice. Two-way ANOVA with Bonferroni's multiple comparisons tests, ***$p < 0.0002$, ****$p < 0.0001$. **c** Male mouse liver transcription profiling by Affymetrix microarray (GEO: GSE55084) shows *Slc46a3* mRNA after treatment with PCB-153 (IP injection, 80 mg/kg for 48 h) and TCDD (IP injection, 10 μg/kg for 48 h). Oil ($n = 6$ per group), PCB-153 ($n = 3$ per group), TCDD ($n = 3$ per group). NS not significant. **d** Putative DRE/XRE binding sites of *Slc46a3* promoter were extracted using ChIP-seq data (GEO accession: GSE97634) and possible enhancer region ($\sim -17{,}135$ bp from the 5'-flanking region of *Slc46a3*) was analyzed using original file sets using Integrated Genome Browser (Bioviz). **e** Counting of putative dioxin-responsive enhancer (DRE)/xenobiotic response element (XRE) region ($-17{,}135$ bp of *Slc46a3*) binding by AhR after TCDD administration in mouse liver ($n = 5$ per group). The ChIP-seq data set was from Gene Expression Omnibus (Geo, accession: GSE97634). **f** Luciferase assay using putative 3×DRE enhancer region ($-17{,}135$ bp) of *Slc46a3* gene in the Hepa1c1c7 cells. Cells were transfected with 3×DRE or 3×mutDRE luciferase constructs ($n = 3$ biologically three independent experiments) overnight and treated with indicated doses of TCDD for 6 h. **g** Inhibitory effect of the AhR inhibitor, CH-223191, on the putative 3×DRE luciferase activity in Hepa1c1c7 cells. After transfection with the 3×DRE luciferase plasmid ($n = 3$ biologically three independent experiments), cells were co-treated with different concentrations of CH-223191 with/without 10 nM TCDD for 6 h. Transfection was normalized with the Renilla luciferase signal. One-way ANOVA with Tukey's multiple comparisons test, ****$p < 0.0001$. **h** Inhibitory effects of the AhR antagonist CH-223191 on TCDD-induced *Slc46a3* mRNA in Hepa1c1c7 cells ($n = 5$ biologically independent samples). **i** Expression of hepatic *Slc46a3*, *Cyp1a2*, and *Ahr* mRNAs as a function of mouse age ($n =$ more than 6 per group). One-way ANOVA with Tukey's multiple comparisons test, ****$p < 0.0001$. **j** Expression of hepatic *Slc46a3*, *Cyp1a2*, and *Ahr* mRNAs as a function of time after TCDD treatment ($n = 4$ per group). One-way ANOVA with Tukey's multiple comparisons test, ****$p < 0.0001$. **k** Expression of *Slc46a3* mRNA as a function of dose and time of oil ($n = 4$ per group) or TCDD ($n = 5$ per group) treatment. Each data point represents the mean ± SD and adjusted $p$ value, presented in the panels, was determined by unpaired two-tailed Student's $t$ test using indicated sample sizes and groups.

lysosome by mCherry-SLC46A3 (Fig. 3f). This might be due to certain metals such as copper, transferred into the lysosome.

**Copper ions may be sequestered in the lysosome** via **SLC46A3**. To probe the lysosomal copper contents, eGFP-SLC46A3-expressing Hepa1c1c7 cells were incubated with the labile copper sensing probe CF4 (1 μM) and its non-responsive control Ctrl-CF4 (1 μM) for 10 m and a red filtered fluorescence signal detected under fluorescence microscopy. The CF4 signal was strongly detected in the eGFP-SLC46A3-embedded lysosomes compared to Ctrl-CF4 (Fig. 3g). By high-magnification images of eGFP-SLC46A and CF4, their sensing location was confirmed using a BZ-X analyzer. eGFP-SLC46A3 protein was localized in

the lysosomal membrane and CF4-Cu was encapsulated inside of the lysosome (Fig. 3h, i). In addition, the lysosome size was increased in eGFP-SLC46A3-expressing cells (Fig. 3j). To measure whether SLC46A3 could induce lysosomal copper accumulation, FLAG-SLC46A-expressing Hepa1c1c7 cells were incubated with CF4. The CF4 signal was significantly increased by FLAG-SLC46A3 (Fig. 3k, l). As a control, the cells were treated with different doses of $CuSO_4$ for 10 min and followed by CF4 (1 μM) treatment for 10 min. Fluorescence intensity of CF4 at 560 nm was significantly increased by $CuSO_4$ in a dose-dependent manner (Fig. 3m). Furthermore, the protein level of the copper chaperone for superoxide dismutase (CCS), a representative marker of the labile copper pool in the cytosol, under different conditions was measured. CCS was significantly increased in both

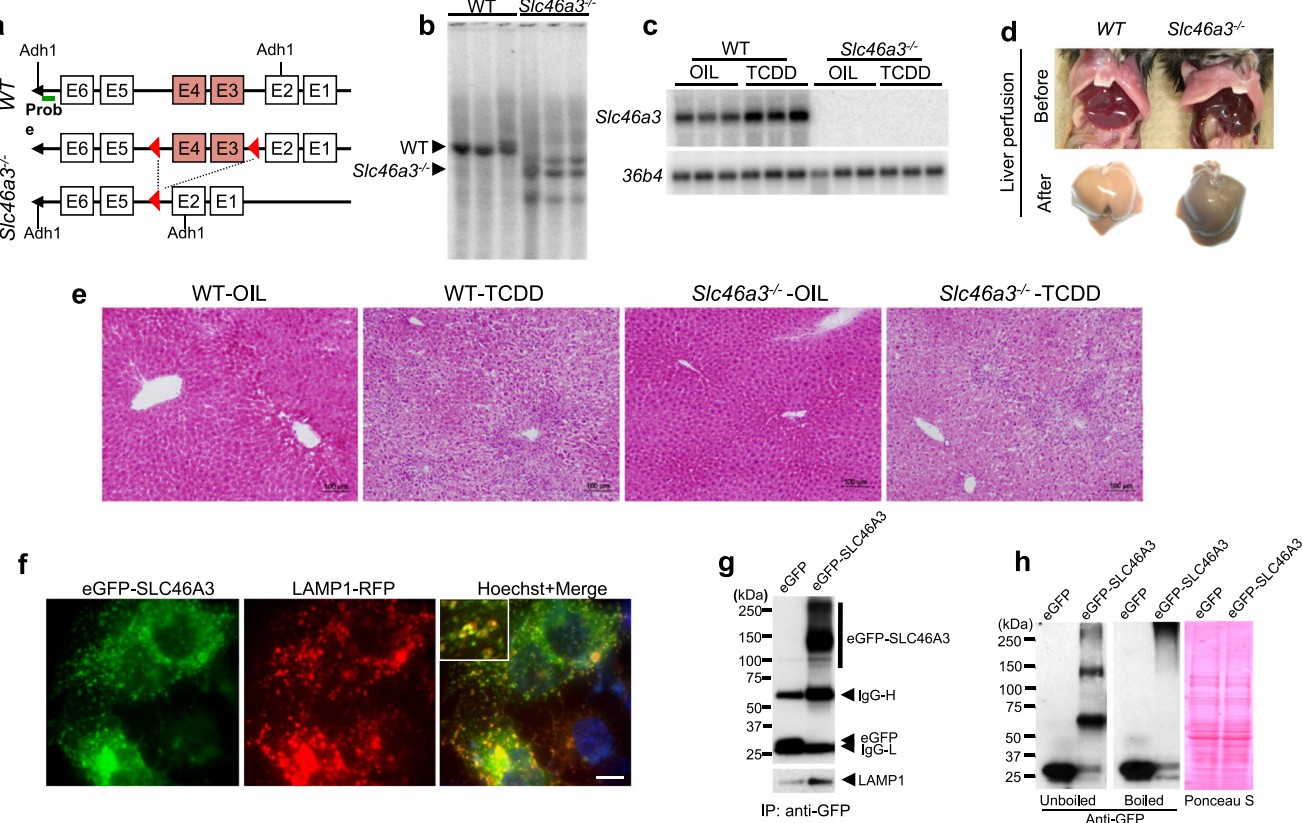

**Fig. 2 Generation and characterization of *Slc46a3*[−/−] mice. a** Scheme for the generation of *Slc46a3*[−/−] mice. **b** Southern blot genotyping of *Slc46a3*[−/−] mice using a specific probe. A representative image is shown from three wild-type (WT) and *Slc46a3*[−/−] mice. **c** Northern blot analysis of *Slc46a3* mRNA in the livers of wild-type (WT) and *Slc46a3*[−/−] mice after treatment with TCDD (200 μg/kg) for 24 h. A representative image is shown from three mice in each treatment group from both mouse lines. **d** Change in the color of the liver before and after perfusion in WT and *Slc46a3*[−/−] mice. **e** Representative H&E staining of liver tissue ($n = 3$ images per mouse). **f** Localization of eGFP-SLC46A3 in the lysosomes of Hepa1c1c7 cells. A representative florescence image was obtained from three biologically independent experiments. Blue color, nucleus; scale bar, 10 μm. **g** Purification of eGFP-SLC46A3-embedded lysosomes. IgG-L immunoglobulin light chain; IgG-H immunoglobulin heavy chain. The representative western blot image was obtained from three independent experiments. **h** Western blotting of eGFP-SLC46A3 using lysates of Hepa1c1c7 cells.

TCDD-treated (10 μg/kg for 7 days) livers (Fig. 4a) and eGFP-SLC46A3-expressing Hepa1c1c7 cells (Fig. 4b). Furthermore, the effect was enhanced in the presence of BCS, a Cu (I) chelator in the media, in the presence of eGFP-SLC46A3 expressing cells (Fig. 4b). However, ferritin-heavy chain, a marker for the labile cytosolic iron pool, was not changed by eGFP-SLC46A3 expression level changes in Hepa1c1c7 cells (Fig. 4c). In addition, mitochondrial SOD activity was increased as reported previously (Fig. 4d)[10]. Inversely, the hepatic CCS level was dramatically decreased by disruption of the *Slc46a3* gene in mice (Fig. 4e). Thus, copper may be a substrate for SLC46A3 in the hepatic lysosome.

**SLC46A3 alters mitochondrial potential and morphology.**
Because copper ion is a cofactor of the complex IV of the respiratory chain, the mitochondrial morphology and mitochondrial potential were analyzed in the presence or absence of SLC46A3. In primary hepatocytes of *Slc46a3*[−/−] mice, mitochondria distal from the nucleus, were thick and elongated compared to small round organelles found in WT mouse liver (Fig. 5a). The mitochondrial area of *Slc46a3*[−/−] mice was significantly larger than in WT hepatocytes (Fig. 5b). The altered mitochondrial morphology was further confirmed by fluorescence microscopy after MitoTracker treatment in mCherry- or mCherry-SLC46A3-expressing primary hepatocytes. In particular, in mcherry-SLC46A3-expressing hepatocytes, mitochondria were transformed to thin and long thread-like

structures associated with weak mitoTracker signals (Fig. 5c). In this experiment, the mCherry signal was not visible because the MitoTracker intensity overwhelms the mCherry signal. In addition, MitoTracker intensity was notably decreased in eGFP-SLC46A3-expressing Hepa1c1c7 cells (Fig. 5d). SLC46A3 induction could inhibit mitochondrial potential in tetramethylrhodamine ethyl ester (TMRE)-treated Hepa1c1c7 cells (Fig. 5e). Furthermore, the initial oxygen consumption rate (OCR) was significantly reduced in eGFP-SLC46A3-expressing Hepa1c1c7 cells (Fig. 5f). As mitochondrial potential was changed by SLC46A3 overexpression, hepatic ATP levels were measured. ATP was significantly decreased by overexpression of eGFP-SLC46A3 in Hepa1c1c7 cells (Fig. 5g). In addition, mitochondrial images were taken using transmission electron microscopy (TEM) to determine whether overexpression of SLC46A3 could change mitochondrial morphology. The mitochondrial morphology in livers of *Slc46a3*[−/−] mice were longer compared to those in WT livers (Fig. 5h). However, a truncated or donut-shaped mitochondrial form was found in livers expressing eGFP-SLC46A3 (Fig. 5i).

***Slc46a3*[−/−] mice are resistant to TCDD-induced hepatic triglyceride accumulation.** To investigate the possible function of hepatic SLC46A3 after TCDD (200 μg/kg for 7 days) exposure, ALT and ALP were measured in serum. However, a difference between WT and *Slc46a3*[−/−] mice was not observed. Interestingly, TCDD did not alter the hepatic triglyceride (TG) levels in *Slc46a3*[−/−] mice

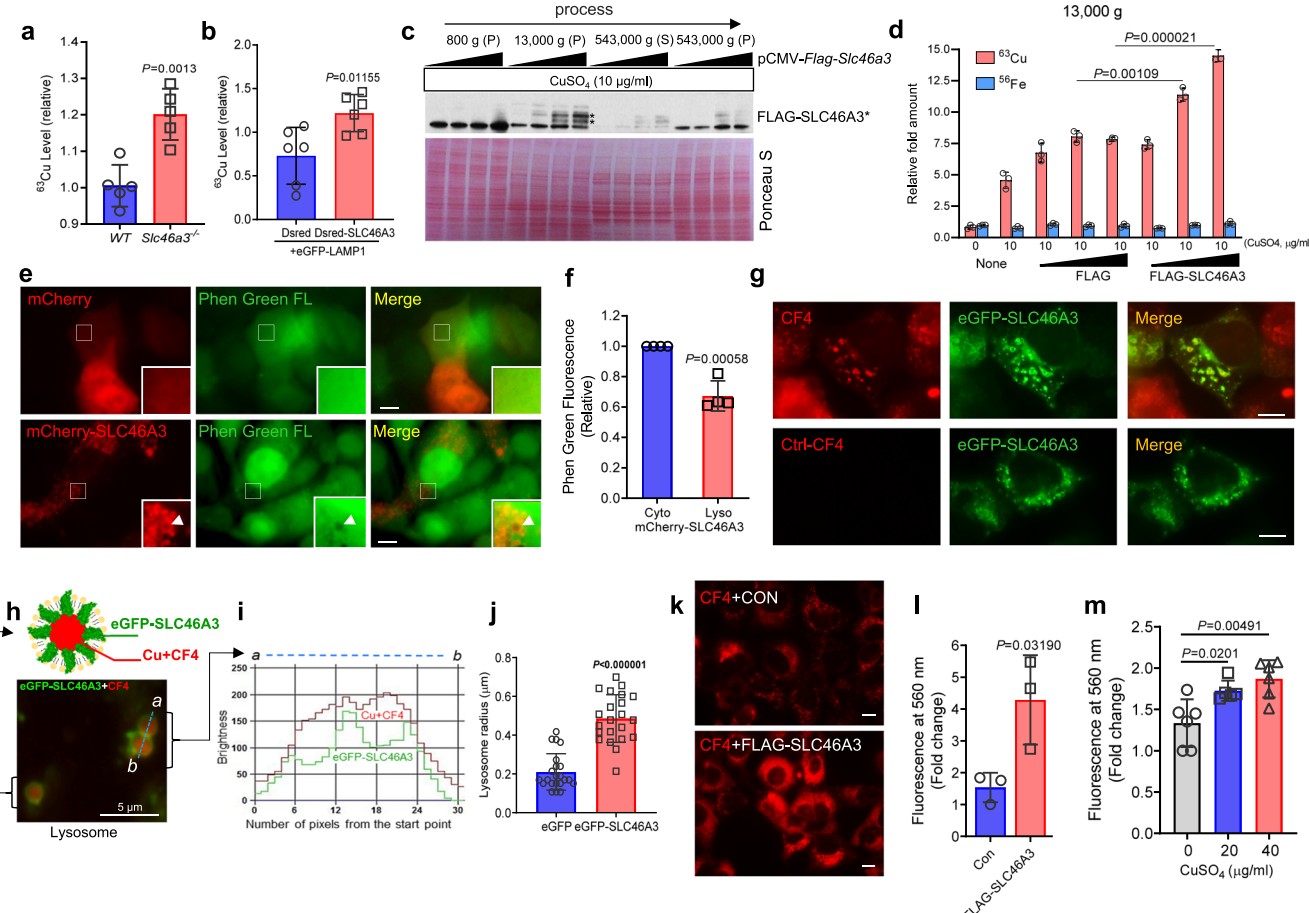

**Fig. 3 Copper is a possible substrate of SLC46A3 in lysosome. a** Hepatic Copper 63 ($^{63}$Cu) level analysis using inductively coupled plasma mass spectrometry (ICP-MS) in WT ($n = 5$ per group) and $Slc46a3^{-/-}$ ($n = 5$ per group) mice. **b** Relative $^{63}$Cu level in the lysosomal fraction from eGFP-Lamp/ DsRed or eGFP-Lamp/DsRed-SLC46A3-expressing mouse liver ($n = 3$ per group). **c** FLAG-SLC46A3 fractions in different centrifugal forces using Hepa1c1c7 cells. P pellet, S supernatant. **d** Copper transporter assay using FLAG-SLC46A3-expressing Hepa1c1c7 cells in the presence of copper sulfate ($n = 3$ biologically independent samples). **e** Measurement of the relative quenching effect by heavy metals using Phen Green FL dye in mcherry-SLC46A3-expressing Hepa1c1c7 cells. The representative images were obtained from three independent experiments. Arrows indicate lysosomes. Scale bar, 10 μm. **f** Relative green fluorescence was measured relative to cytoplasmic and lysosomal locations. Cyto cytoplasm, Lyso lysosome. **g** Fluorescence images show labile copper-sensing probe CF4 localized in the eGFP-SLC46A3 embedding lysosomes in Hepa1c1c7 cells. Ctrl-CF4 probe was used as a control. All images were pictured under the same conditions. The representative images were obtained from three independent experiments. Scale bar, 10 μm. **h** A model of copper-encapsulated lysosome with eGFP-SLC46A3 was depicted. The representative images were obtained from three independent experiments. **i** Locations of eGFP-SLC46A3 and Cu-CF4 were presented with the brightness of each color from the section (a to b) of two neighboring lysosomes. **j** Lysosomal sizes were measured in the presence of eGFP or eGFP-SLC46A3 in Hepa1c1c7 cells. **k** Hepa1c1c7 cells were transfected with FLAG-SLC46A and incubated with CF4 (1 μM) for 10 m. Red fluorescence color indicates Cu-CF4 complex. The representative images were obtained from three independent experiments. Scale bar, 10 μm. **l** The Red-light intensity was measured using a fluorescence plate reader (Ex. 540/Em 560). **m** To control the CF4-Cu sensing experiment, Hepa1c1c7 cells were treated with different doses of CuSO$_4$ for 10 m and incubated with CF4 (1 μM) for 10 m. The CF4 emission light was read using a fluorescence plate reader (Ex. 540/Em 560). Each data point represents the mean ± SD and adjusted p value, presented in the panels, was determined by unpaired two-tailed Student's t test using indicated sample sizes and groups.

compared to WT mice. In addition, differences in levels of hepatic free fatty acids (non-esterified fatty acid, NEFA) were not observed between both groups in the presence or absence of TCDD (Fig. 6a). Oil red O staining revealed that lipid droplet accumulation in livers of TCDD-treated $Slc46a3^{-/-}$ mice were significantly reduced compared to WT mice (Fig. 6b). Since $Slc46a3^{-/-}$ mice were resistant to TCDD-induced TG accumulation in the liver, a high-fat diet (HFD) model was applied to determine whether $Slc46a3^{-/-}$ mice were resistant to hepatic TG accumulation. Notably, after 3 weeks of HFD feeding, body weight changes in $Slc46a3^{-/-}$ mice were significantly reduced compared to WT mice (Fig. 6c). Glucose tolerance test (GTT) showed no significant difference between WT and $Slc46a3^{-/-}$ mice (Supplementary Fig. 3). In addition, liver mass

and epididymal fat mass were also lower in $Slc46a3^{-/-}$ than in WT livers (Fig. 6d, e). Hepatic TG levels were lower in $Slc46a3^{-/-}$ after HFD feeding compared to WT mice. Conversely, hepatic NEFA levels were higher in $Slc46a3^{-/-}$ mice than in WT mice (Fig. 6f), while differences in serum TG and NEFA levels in the two groups were not observed (Fig. 6g). These results suggest that SLC46A3 influences hepatic TG generation from NEFA in the liver. Due to the change of hepatic TG levels in $Slc46a3^{-/-}$ mice, lipid metabolites were analyzed in livers from WT and $Slc46a3^{-/-}$ mice. Two peaks from 16.88 min were significantly changed (Supplementary Fig. 4a). PLS-DA (Supplementary Fig. 4b) and S-plots (Supplementary Fig. 4c) using UPLC-MS data by SIMCA-P + 12 software, as well as GC-MS analysis with the FAME-derivatization method,

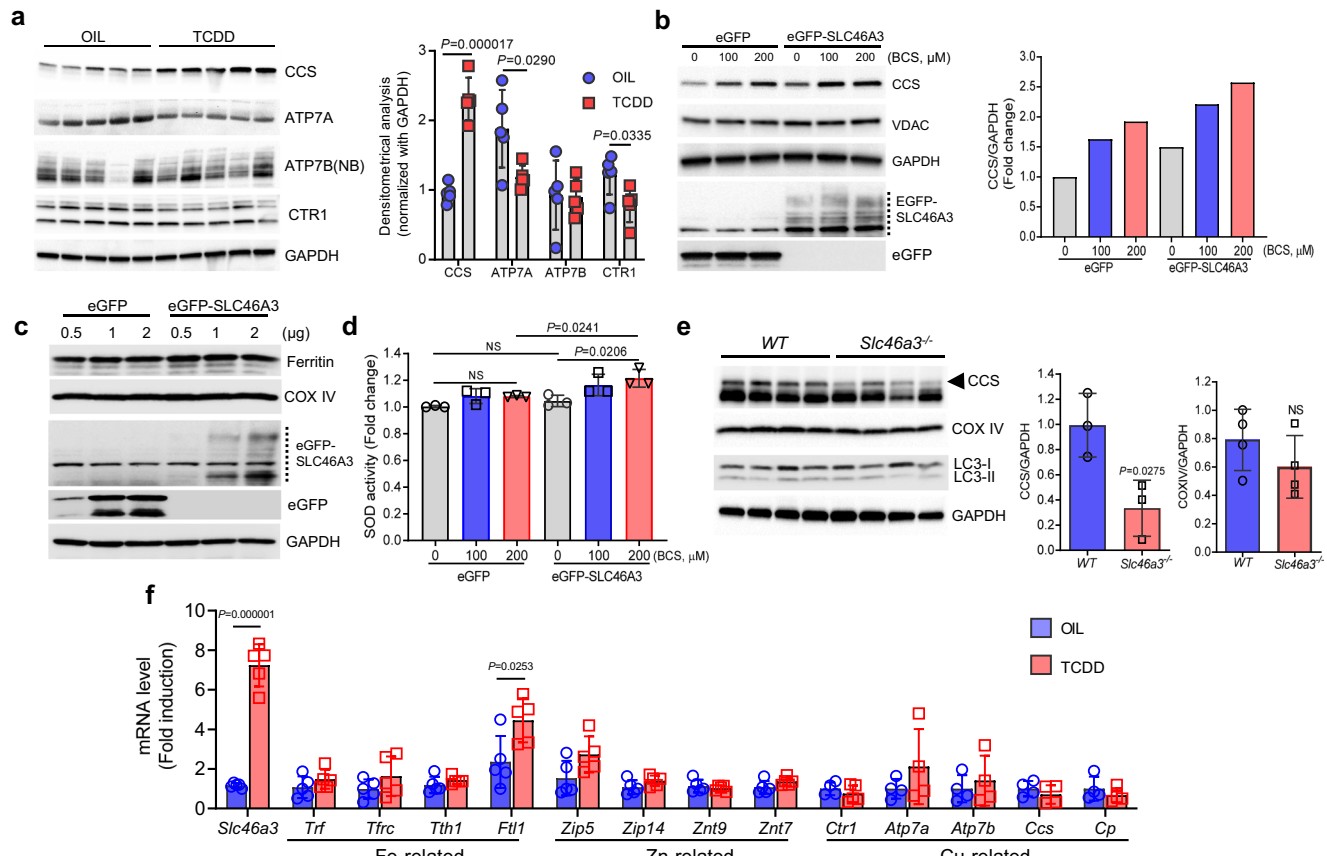

**Fig. 4 TCDD and eGFP-SLC46A3 increases the copper chaperone for superoxide dismutase. a** Western blotting data show the level of copper-related proteins in the liver after oil ($n = 5$ per group) or TCDD (10 μg/kg for 7 days) treatment ($n = 5$ per group). Densitometrical analysis is shown in the right panel. **b** Western blotting data show the level of copper chaperone for superoxide dismutase (CCS) protein level after treatment with BCS (Cu (I) chelator in the media) for 3 days in the presence of eGFP-SLC46A3 in Hepa1c1c cells. Densitometrical analysis is shown in the right panel. NB nonboiled sample. **c** The level of heavy chain of ferritin, an indicator of labile cytoplasmic iron levels, was measured after induction of eGFP-SLC46A3 in Hepa1c1c7 cells. **d** SOD activity was measured when Hepa1c1c7 cells were treated with different doses of BCS in the presence of eGFP-SLC46A3. **e** Western blotting data show the level of copper chaperone for superoxide dismutase (CCS) in the livers from wild-type (WT)($n = 4$ per group) and $Slc46a3^{-/-}$ mice ($n = 4$ per group). Densitometrical analysis is shown in the right panel. **f** Expression of mRNAs encoding metal transporters in the mice liver after oil ($n = 4$ per group) or TCDD (10 μg/kg for 24 h) ($n = 5$ per group) administration. Each data point represents the mean ± SD and adjusted $p$ value, presented in the panels, was determined by unpaired two-tailed Student's $t$ test using indicated sample sizes and groups.

revealed two ions [16:0/18:1/18:2] and TG [16:0/18:2/ 18:2] as TG (Supplementary Fig. 4d). Thus, hepatic levels of *Acadm, Acox1, Fasn, Acc1, Scd1, Lcat, Cd36, Hsl, Pnpla2, Aadac, Tgh, Mttp, Dgat1, Dgat2, Fsp27,* and *Plin2* mRNAs encoding TG/NEFA-related proteins were measured. However, no difference between WT and $Slc46a3^{-/-}$ in their expression levels was found under the condition of both vehicle and TCDD treatment (Fig. 6h).

***Slc46a3* induction triggers hepatic triglyceride accumulation**. Since the phenotype of the $Slc46a3^{-/-}$ mice revealed that SLC46A3 affected hepatic TG accumulation, a direct role for SLC46A3 in lipid accumulation was investigated. Forced expression of SLC46A3 in the liver by both adenovirus and hydrodynamic sheer injection significantly increased hepatic lipids (Fig. 7a) and TGs (Fig. 7b). Since adipophilin (ADFP) is expressed in lipid droplets in concert with TG accumulation, *Adfp* mRNA levels were measured after Ad-SLC46A3 administration for 7 days. Hepatic *Adfp* mRNA was significantly increased by Ad-SLC46A3 administration (Fig. 7c). In addition, mCherry-SLC46A3 increased the size of eGFP-ADFP-expressing lipid droplets in primary hepatocytes (Fig. 7d, e). The increase of lipid droplet size by SLC46A3 was confirmed by Nile Red staining in primary

hepatocytes (Fig. 7f). In addition, lipid droplets were increased by tetraethylenepentamine (TEPT), a copper chelator, in Hepa1c1c7 cells (Supplementary Fig. 5). Furthermore, SLC46A3 could alter the phosphorylation of some energy homeostasis-related proteins such as APMKα and ACC. Ad-SLC46A3 increased phosphorylation of AMPKα and decreased phosphorylation of ACC in a titer-dependent manner in primary hepatocytes (Fig. 7g). Thus, SLC46A3 induction may cause unbalanced energy homeostasis.

## Discussion

Here, the function of SLC46A3 was analyzed in mouse liver. Because SLC46A3 is specifically induced by TCDD in liver via AhR signaling, it might play an important role in liver pathophysiology, notably in the toxic response to TCDD. To investigate the function of SLC46A3, $Slc46a3^{-/-}$ mice were generated. The critical gross pathological feature of $Slc46a3^{-/-}$ mice was gray rather than yellow livers as compared to WT mice. Metal ions were measured in the liver to investigate whether the color change was the result of altered levels of intracellular metal ions. Interestingly, copper and iron were increased in the liver. For iron, no significant difference was noted after liver perfusion as well as in the 13,000 g fraction of the eGFP-SLC46A3-expressing liver

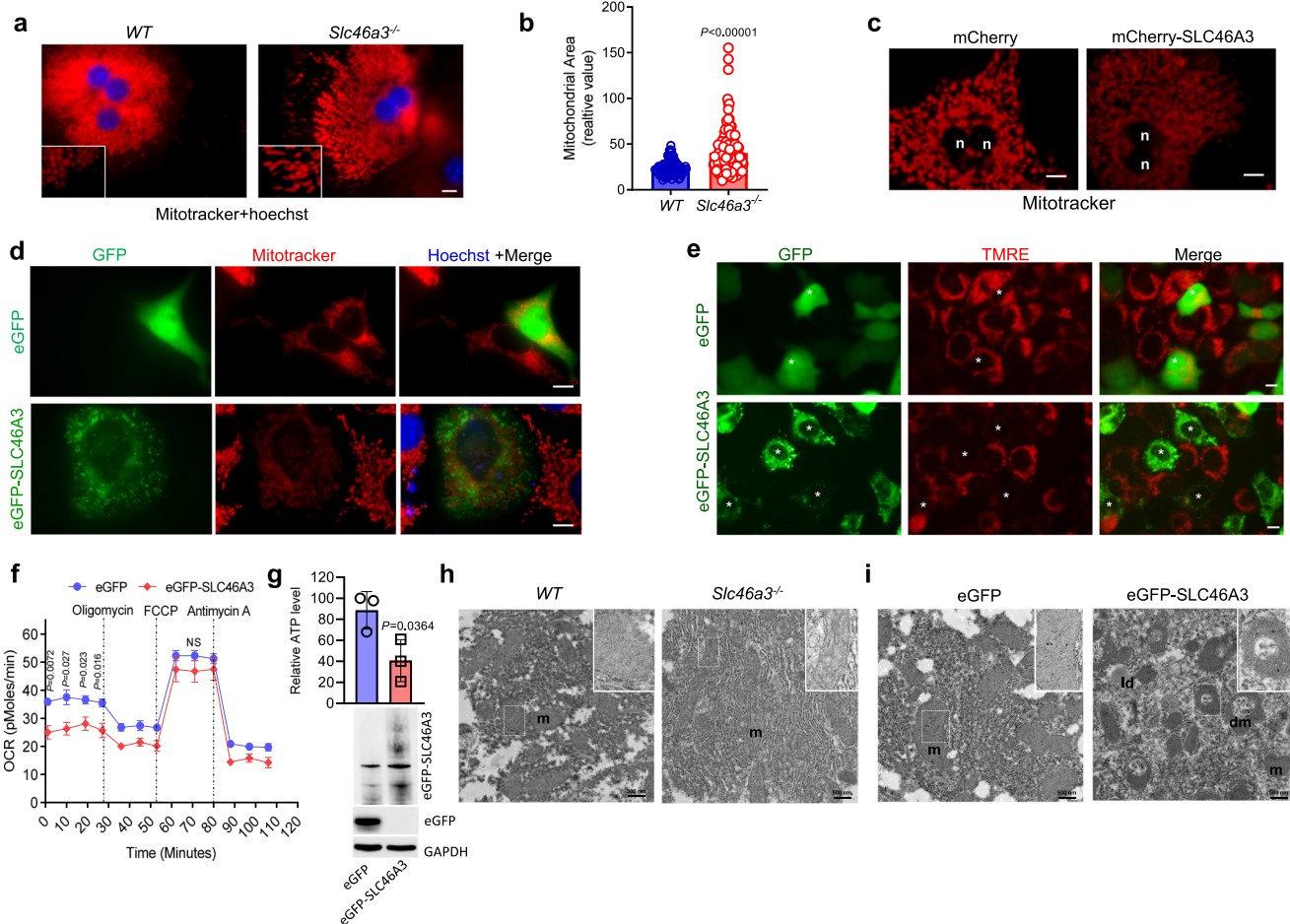

**Fig. 5 Slc46a3 affects mitochondrial potential and morphology. a** Fluorescence images of mitoTracker in the primary hepatocytes. Red color, mitochondria; blue color, nucleus; scale bar, 10 μm. The representative images were obtained from three independent experiments. **b** Relative mitochondrial sizes between WT and $Slc46a3^{-/-}$. **c** MitoTracker intensity in the primary hepatocytes in the presence or absence of mCherry-SLC46A3. The representative images were obtained from three independent experiments. N nucleus; scale bar, 10 μm. **d** MitoTracker intensity in the eGFP-SLC46A3-expressing Hepa1c1c7 cells. Scale bar, 10 μm; blue color, nucleus. The representative images were obtained from three independent experiments. **e** Mitochondrial potential images with TMRE dye in Hepa1c1c7 cells. The asterisk indicates the same location. The representative images were obtained from three independent experiments. **f** Mitochondrial oxygen consumption rate (OCR) in eGFP-SLC46A3-expressing Hepa1c1c1 cells. **g** Relative ATP levels were measured in the eGFP-SLC46A3-expressing Hep1c1c7 cells. **h** Transmission electron microscopy (TEM) images of hepatocytes in WT ($n = 3$ per group) and $Slc46a3^{-/-}$ ($n = 3$ per group) mice. m mitochondria; scale bar, 500 nm. The representative images were obtained from three biologically independent liver samples. **i** TEM images of hepatocytes in eGFP-SLC46A3-expressing liver. The representative images were obtained from three biologically independent liver samples. ld lipid droplet, dm donut shape mitochondria, m mitochondria; scale bar, 500 nm. Each data point represents the mean ± SD and the adjusted $p$ value, presented in the panels, was determined by unpaired two-tailed Student's $t$ test using indicated sample sizes and groups.

(Supplementary Fig. 6). This may be due to an increase in iron in the blood. In this study, the relationship between copper ion and SLC46A3 in the liver was investigated. Because SLC46A3 was localized in the lysosome, the isolated lysosome fraction was subjected to metal analysis using inductively coupled plasma mass spectrometry (ICP-MS). These results suggested that copper could be a possible substrate of SLC463A3.

Lysosomes are catabolic organelles in eukaryotic cells that digest various components from the cytoplasm. Previous reports suggested that copper can be transported to hepatocytes through vesicle pathways and can be excreted into the bile by lysosomes[11,12]. Lysosomes were enlarged by SLC46A3 expression, however, the mechanism is not known. SLC46A3 may influence the recruitment of functional proteins for lysosomal activation such as LAMP1. Misregulated or mutant lysosomal membrane protein-coding genes can cause human diseases. For example, a mutation in *MCOLN1* encoding mucolipin-1, a lysosomal cation channel for ions such as $Na^+$, $K^+$, $Ca^{2+}$, $Fe^{2+}$,

causes mucolipidosis type IV[13]. As SLC46A3 protein could modulate lysosomal copper, unbalanced intracellular copper levels in lysosomes can lead to altered metabolic pathways as well as other metal-related transporting systems in the liver. Increased copper levels as a result of TCDD administration or SLC46A3 overexpression/deletion could influence the expression of copper-related genes or other metal concentrations in the cells. Iron-related gene mRNAs (*Trf*, *Tfrc*, *Tth1*, and *Ftl1*), zinc-related gene mRNAs (*Zip5*, *Zip14*, *Znt9*, and *Zint7*), and copper-related gene mRNAs (*Ctr1*, *Atp7a*, *ATp7b*, *Ccs*, and *Cp*) were measured and only *Ftl1* mRNA was slightly increased by TCDD in the liver. This suggests that copper deficiency may trigger the increase of iron level as a compensatory mechanism. However, hepatic mRNA levels of zinc or copper-related proteins were unchanged after TCDD administration.

Copper is an essential transition metal that acts as a cofactor in many enzymes and plays an important role in cellular metabolism and bioenergy. Importantly, copper ion is a cofactor for

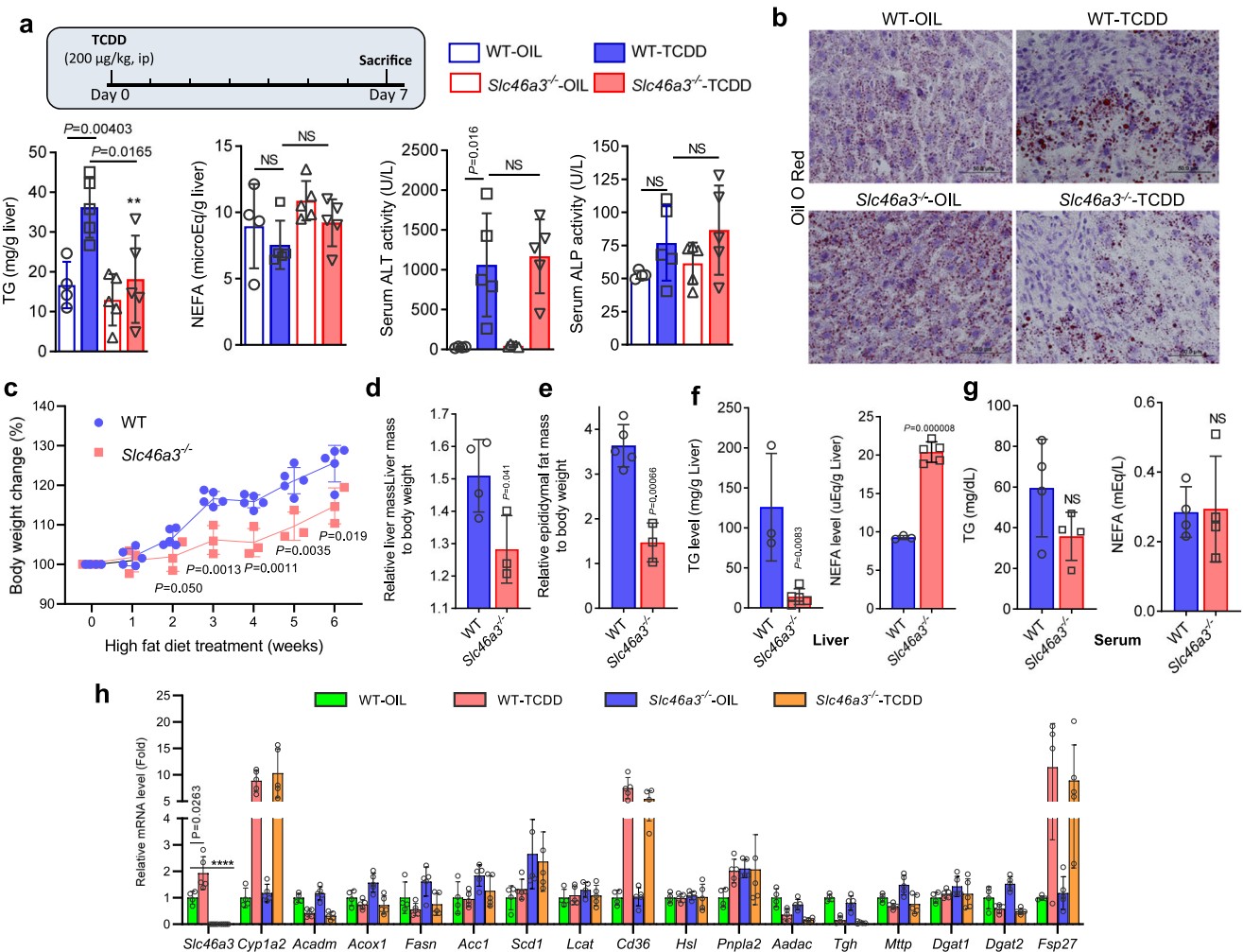

**Fig. 6 TCDD induces accumulation of hepatic triglyceride through SLC46A3. a** Measurements of hepatic triglyceride (TG) and non-esterified fatty acid (NEFA) as well as serum ALT activity and serum ALP activity (*n* = 4 for WT-OIL group; *n* = 5 for other groups). Two-way ANOVA with Bonferroni's multiple comparisons test, **\*\****p* = 0.0062; NS not significant. **b** Representative Oil O Red staining of liver tissue. **c** Body weight changes after high-fat diet WT (*n* = 4 per group) and *Slc46a3*⁻/⁻ (*n* = 3 per group) mice. **d** Liver mass to body weight after high-fat diet in WT (*n* = 4 per group) and *Slc46a3*⁻/⁻ (*n* = 3 per group) mice. **e** Epididymal fat mass after high-fat diet in WT (*n* = 5 per group) and *Slc46a3*⁻/⁻ (*n* = 3 per group) mice. **f** Hepatic TG and NEFA levels after high-fat diet in WT (*n* = 3 per group) and *Slc46a3*⁻/⁻ (*n* = 5 per group) mice. **g** TG and NEFA levels in the serum after high-fat diet in WT (*n* = 4 per group) and *Slc46a3*⁻/⁻ (*n* = 4 per group) mice. NS not significant. **h** Hepatic mRNA expressions of lipid-related metabolizing genes by oil or TCDD (10 μg/kg for 24 h) in WT and *Slc46a3*⁻/⁻ mice. Two-way ANOVA with Bonferroni's multiple comparisons test, **\*\*\*\****p* < 0.0001. The sample sizes of WT-OIL, WT-TCDD *Slc46a3*⁻/⁻-OIL, and *Slc46a3*⁻/⁻-TCDD are *n* = 4, 5, 5, 5 mice per group, respectively. Each data point represents the mean ± SD and the adjusted *p* value, presented in the panels, was determined by unpaired two-tailed Student's *t* test using indicated sample sizes and groups.

mitochondrial respiratory complex IV and cytochrome c oxidase, and thus intracellular copper deficiency may cause mitochondrial dysfunction[14–17]. Features of impaired copper metabolism lead to genetic disorders such as Wilson's and Menkes disease[14–16,18]. Moreover, copper regulates the lipolysis of triglycerides via cAMP signaling[19]. Therefore, in the absence of SLC46A3, the levels of copper in the cells could be increased and mitochondrial function would be increased as well. Conversely, induction of SLC46A3 could lead to copper deficiency, since copper could migrate to the lysosome for excretion. Indeed, previous results show that hepatic copper deficiency appears at the early stages of HFD feeding[20]. According to a previous report[21], ATP7B, which results in Wilson's disease, removes excess copper into the bile via lysosomal exocytosis. Thus, lysosomal SLC46A3 trafficking is critical to understand copper deficiency in hepatocytes. As a conventional method for detecting copper accumulation in the liver, rhodanine staining was used and copper staining was noted in *Slc46a3*⁻/⁻ mice. Because copper was

increased by about 30% in the liver of *Slc46a3*⁻/⁻ mice, copper levels were measured in isolated lysosomes using ICP-MS analysis. Copper accumulation was also measured in SLC46A3-embedded lysosomes by using the copper sensing probe CF4. Based on this study, the characteristics of mitochondria differed according to the levels of SLC46A3. As expected, intracellular copper deficiency resulted in inhibition of mitochondrial potential when SLC46A3 was over-expressed. This result is supported by a previous report that mito-chondrial potential is significantly reduced by TCDD in rat hepatocytes[22]. In addition, in a previous, study the morphology of some mitochondria was changed by copper deficiency to a donut shape in human hepatocytes[23] as well as the thymus and spleen in mice[24]. In addition, others reported that mitochondria were changed to a donut shape by carbonyl cyanide m-chlorophenyl hydrazine (CCCP), a mitochondrial oxidative phosphorylation uncoupler, without fusion or fission in MEF cells[25]. This morphological feature was similar to the present result in a fusion- or fission-independent

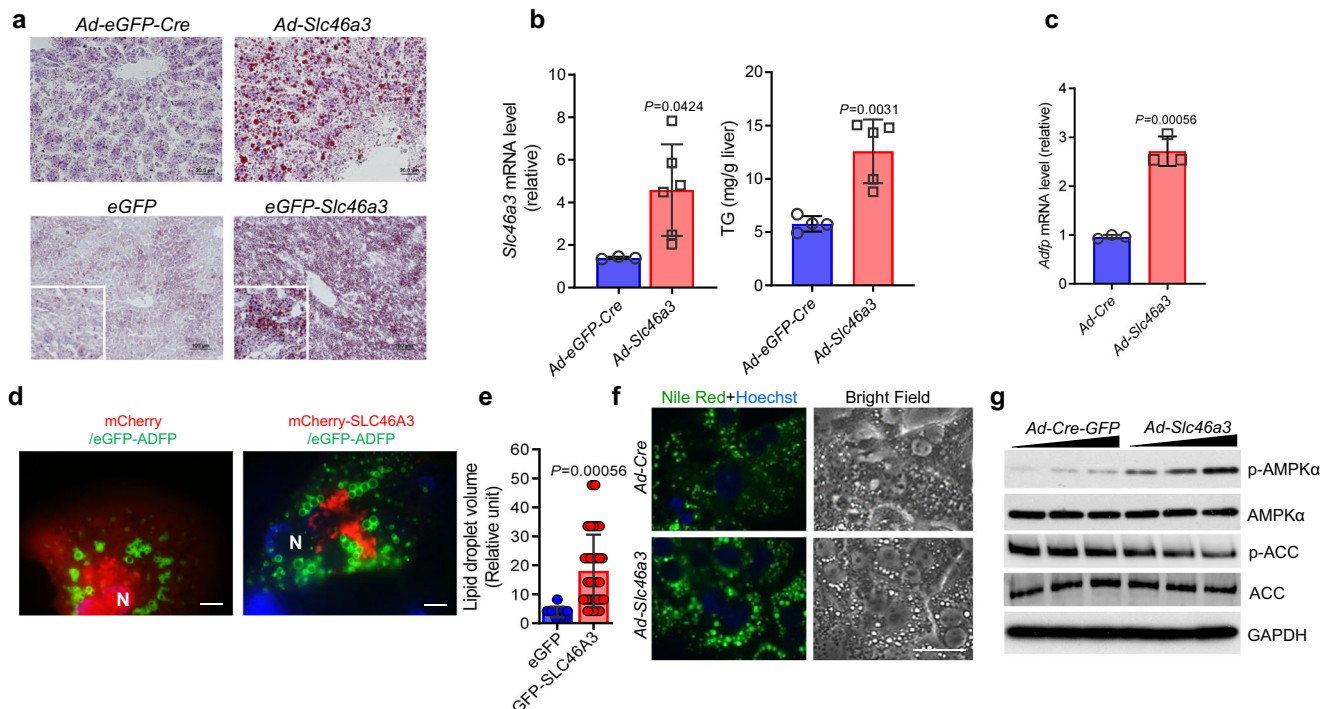

**Fig. 7 SLC46A3 overexpression triggers TG accumulation in the liver. a** Oil Red O staining after adenovirus-based (upper, $n = 5$ per group) and hydrodynamic injection-based (lower, $n = 3$ per group) expression of *Slc46a3* mRNA in the liver. **b** Hepatic *Slc46a3* mRNA expression (left) and TG level (right) after administration of *Ad-eGFP-Cre* ($n = 3$ mice per group) or *Ad-Slc46a3* ($n = 5$–6 mice per group). **c** Hepatic *Adfp* mRNA expression after *Ad-Slc46a3* and control *Ad-Cre* administration ($n = 3$ per each group). **d** Representative fluorescence images in the eGFP-adipophilin/mcherry-SLC46A3-expressing primary hepatocytes. The representative images were obtained from three biologically independent experiments. Scale bar, 10 μm; N nucleus (blue color). **e** Relative volume of lipid droplets in the eGFP ($n = 11$ counts per cell) or eGFP-SLC46A3 ($n = 32$ counts per cell) expressing primary hepatocytes from (**d**). **f** Nile Red staining after adenoviral induction of SCL46A3 in primary hepatocytes with *Ad-Slc46a3*. The representative fluorescence images were obtained from three biologically independent experiments. Scale bar, 10 μm. **g** Western blotting of phosphorylated AMPKα and ACC using the mice liver after viral induction of *Ad-Cre-GFP* or *Ad-Cre-Slc46a3*. The representative western blot images were obtained from three biologically independent experiments. Each data point represents the mean ± SD and the adjusted *p* value, presented in the panels, was determined by unpaired two-tailed Student's *t* test using indicated sample sizes and groups.

manner by the level of SLC46A3. Thus, SLC46A-induced copper deficiency might lead to mitochondrial dysfunction. Later, TG could accumulate in the liver after the reduction in mitochondrial function. In addition, copper may affect hepatic autophagy[26]. However, changes of LC3 were not found in the absence of SLC46A3.

High TG has been linked to atherosclerosis and heart disease in humans. Under normal circumstances, an excessive amount of TG can be stored in the form of lipid droplets and later used as an energy source[27]. However, abnormal TG accumulation caused by SLC46A3-induced mitochondrial dysfunction can cause serious pathophysiological problems. In a previous study, hepatic TG levels were dramatically increased by copper deficiency in rats[28]. Therefore, intracellular copper levels could play an important role in lipid metabolism in conjunction with mitochondria function. TCDD treatment-induced mRNA expression of some TG/NEFA-related proteins. However, the induction was not SLC46A3-dependent, and could not explain the SLC46A3-dependent TG accumulation in the liver. Induction of hepatic SLC46A3 increased phosphorylation of AMPKα and decreased phosphorylation of ACC. These data indicate the SLC46A3 can cause mitochondrial dysfunction and energy deficiency in hepatocytes. In conclusion, TG accumulation by TCDD is induced by the induction of lysosome SLC46A3, which causes mitochondrial dysfunction by intracellular copper deficiency leading to hepatic steatosis (Fig. 8).

## Methods

**Chemicals**. TCDD was purchased from Cambridge Isotope (Andover, MA). Anti-phospho-AMPKα (#2535), anti-AMPK (#2532), anti-phospho-ACC (#11818),

anti-ACC (#3676), anti-VDAC (#4661 T), anti-COX IV (#4850), anti-Ferritin (#sc-376594), anti-LC3 (#2775), anti-OPA1(#80471), and anti-mouse IgG, HRP-linked secondary antibody (#7076 S), anti-rabbit IgG, and HRP-linked secondary antibody (#7074 S) were obtained from Cell Signaling Technology (Beverly, MA, USA). Anti-GAPDH (#sc-47724), anti-CCS (#sc-374205), anti-Drp1 (#sc-271583), and anti-GFP antibody (#sc-9996) were purchased from Santa Cruz Biotechnology (Santa Cruz, CA, USA). Anti-FLAG antibody (#F1804), bathocuproinedisulfonic acid (BCS), and other chemicals were obtained from Sigma-Aldrich (St. Louis, MO). Nile Red, Mitotracker, and Phen Green FL were purchased from Invitrogen (Carlsbad, CA). JetPEI transfection reagent was obtained from VWR International (Radnor, PA, USA). Anti-Mitofusin2 (#ab56889), TMRE, and superoxide dismutase (SOD) kit were purchased from Abcam (Cambridge, MA). CH-223191 was obtained from Tocris (Bristol, UK). TransIT-QR Hydrodynamic Delivery Solution was purchased from Mirus (Pittsburgh, PA, USA). EZ-ATP Assay kit was obtained from DoGenBio (Seoul, Korea). All other chemicals were of the highest grade commercially available. CF4 and Ctrl-CF4 were synthesized as previously reported[29].

**Animals and treatments**. All animal experiments were conducted in accordance with the Association for Assessment and Accreditation of Laboratory Animal Care international guidelines and approved by the US National Cancer Institute Animal Care and Use Committee. *Slc46a3*$^{-/-}$ mice were generated by disruption of exons 3 and 4 and interbred with heterozygotes to obtain wild-type (WT) littermate (C57BL/6NCr and FVB/N mixed background). *Ahr*$^{-/-}$ (ref. [30]), *Arnt*$^{fl/fl}$ (ref. [31]) mice were described in earlier papers. *Arnt*$^{ΔLiv}$ mice were obtained by crossing the *Arnt*$^{fl/fl}$ mice with mice carrying the Cre transgene under the control of the rat serum albumin gene promoter[32,33]. C57BL/6 J mice were obtained from the Jackson Laboratory (Bar Harbor, ME). All of the mice were maintained under a standard 12 h light/dark cycle with water and chow provided ad libitum. For genotyping of *Slc46a3*$^{-/-}$ mice, mouse tails were subjected to general PCR using specifically designed primers: FJG99-1stloxp-F3, 5′-CC TCGAGGGACCTAATAACTTC-3′, FJG99-15030-F, 5′-GCACTGAGACA CCATTGTGACGCC-3′, and FJG99-15514-R, 5′-AGCAAAGGTCCGCTTAGTTA GAGAC-3′. A PCR amplicon (~400 bp) was indicated as *Slc46a3*$^{-/-}$ mice. To investigate the potential role of SLC46A3 in TCDD-induced liver injury, mice were injected

**Fig. 8 Scheme showing a possible role of SLC46A3 in hepatic lipid accumulation in mice.** In the normal state of hepatocytes, intracellular copper levels can be balanced by homeostasis, but induction of SLC46A3 by TCDD can lead to an abnormal increase in copper migration to lysosomes. Conversely, a mild increase of copper in the cytosol may inhibit TG accumulation in liver. Thus, the lack of intracellular copper levels can result in mitochondrial dysfunction. Ultimately, lipids such as TG are stored in the form of lipid droplets. Nu Nucleus; Mito Mitochondria; Ly Lysosome; SLC SLC46A3; Cu Copper; Ld Lipid droplet; Bc Bile canaliculi.

intraperitoneally with a single dose of 10 or 200 μg/kg of TCDD for 3 or 7 days. To investigate a role for SLC46A3 in hepatic TG accumulation, mice were fed a 60% of HFD that was obtained from Research Diets (New Brunswick, NJ). Six weeks later, the mice were killed.

**Cell culture.** Hepa1c1c7 cells were purchased from the ATCC and cultured in DMEM supplemented with 10% heat-inactivated FBS and 50 U/ml of penicillin/streptomycin mixture (Invitrogen) at 37°C in a humidified atmosphere of 5% $CO_2$/95% air. Cells were grown to 60–80% confluence and trypsinized with 0.05% trypsin containing 2 mM EDTA. For ectopic expression of proteins in hepatocytes, recombinant adenovirus or hydrodynamic shear plasmid DNA was delivered to the liver using tail vein injection. Primary hepatocytes were cultured overnight and treated with 20 nM MitoTracker or 0.5 μg/ml Nile Red for 15–30 m.

**Plasmids.** For mammalian expression, coding regions for mouse *Slc46a3*, *Adfp*, and *Lamp1* cDNAs were subcloned into pEGFP, pDsred, and pmCherry plasmids (Clontech, Mountain View, CA, USA) for tagging eGFP, Dsred, and mCherry, respectively, at the N-terminal of the SLC46A3 protein. For the FLAG-SLC46A3-expressing plasmid (pCMV-Flag-Slc46a3), the EGFP cDNA in the pEGFP plasmid was replaced to make the final Flag-Slc46a3 fusion gene by modification.

**In vivo and in vitro DNA delivery.** For in vivo plasmid DNA delivery, hydrodynamic shear tail vein injection was carried out using 10-week-old C57BL/6 J male mice. Mixtures with pEGFP-Slc46a3 (20 μg) or pmCherry-Slc46a3 (20 μg) or pEGFP-Adipophilin (10 μg) and 2.2 ml of TransIT-QR delivery solution were injected within 3 to 5 s through the tail vein. One day later, the mice were killed or primary hepatocytes isolated for culturing as previously described[34]. Using small pieces of fresh liver, the expression of each fluorescence fusion protein was monitored under fluorescence microscopy. Adenoviral delivery of SLC46A3 using 10-week-old C57BL/6 J male mice was carried out by injecting 200 μl of saline solution containing $2 \times 10^{10}$ pfu/mouse through the tail vein. A week later, the mice were killed. For in vitro transfection, Hepa1c1c7 cells were transfected with different plasmids using JetPEI reagent for 16 h before the cells were harvested.

**Quantitation of mRNA.** Total RNA was isolated from the liver using Trizol (Invitrogen, Carlsbad, CA). After synthesis of complementary DNA (cDNA) using a SuperScript II reverse transcriptase kit (Invitrogen, Carlsbad, CA), qPCR was carried out using an Applied Biosystems Prism 7900HT Sequence Detection System (Foster City, CA) as described previously[35]. Expression levels of mRNA were normalized to *18 S* RNA or *Gapdh* mRNA as the internal standards. Primers for the qPCR are listed in Supplementary Table 1.

**Western blotting.** Hepa1c1c7 cells were lysed with M-PER buffer (Thermo Scientific) or RIPA lysis buffer (150 mM NaCl, 0.5% Triton X-100, 50 mM Tris–HCl, pH 7.4, 25 mM NaF, 20 mM EGTA, 1 mM DTT, 1 mM $Na_3VO_4$, and protease inhibitor cocktail) for 30 min on ice, followed by differential centrifugal for fractionation. Protein concentration was measured with the bicinchoninic acid (BCA) reagent. Protein (10–30 μg), denatured by boiling and unboiled control, was subjected to electrophoresis on a 4–15% gradient Tris-HCl gel (Bio-Rad, Hercules, CA) and then electrotransferred onto a polyvinylidene difluoride membrane in Tris-glycine buffer (pH 8.4) containing 20% methanol. The membrane was blocked with 5% fat-free dry milk in phosphate-buffered saline containing 0.1% Tween-20 (PBST) for 1 h. The membranes were probed with primary antibodies and horseradish peroxidase-conjugated secondary antibodies (#7074 S, #7076 S, Cell Signaling) using standard western blotting procedures. Secondary antibodies were diluted at 1:5000 for the experiments. Proteins were visualized using the Femto signal chemiluminescent substrate (Pierce) by an image analyzer (Alpha Innotech Corp., San Leandro, CA) or a Chemidoc imaging system (Bio-Rad). Protein densitometry analyses were performed using Image Lab 6.1.1 software (Biorad).

**Mitochondria analysis.** For mitochondrial OCR measurements, Hepa1c1c7 cells were cultured in specialized 24-well plates and transfected with 100 μg of pEGFP or pEGFP-Slc46a3 plasmid for 24 h and mitochondrial OCR measured using the Seahorse XF24 analyzer as described previously[36]. OCR was measured from the basal conditions and later cells were treated with the mitochondrial inhibitor oligomycin, a mitochondrial uncoupling compound carbonylcyanide-4-trifluoromethoxyphenylhydrazone (FCCP), and the respiratory chain inhibitor antimycin A.

**Biochemical analysis.** Serum levels of alanine aminotransferase (ALT) and alkaline phosphatase (ALP) were measured using ALT (Catachem, Bridgeport) and ALP kits (Catachem, Bridgeport). Serum or hepatic level of non-esterified fatty acids (NEFA) and hepatic TG level were quantified using TG kit (Wako Chemicals USA Inc.) and NEFA kit (Wako Chemicals USA Inc.) according to the manufacturer's instructions.

**Metal analysis in the liver.** Liver tissues and isolated cell fractions were treated with 100% nitric acid at 70 °C for 3 h and then incubated overnight at room temperature. Levels of major physiological metal ions were measured by inductively-coupled plasma (ICP)-MS (Agilent, Model 7500 cs) as described previously[37].

**Copper transport assay.** To analyze the copper levels in the lysosomal fraction, hepatic lysosomal fraction was isolated using the magnetic bead method. In detail, 8-week-old FVB male mice were transfected with pEGFP-Lamp1 (10 μg) and pDsred-Slc46a3 (10 μg) or pDsred (10 μg) using the hydrodynamic injection method for 24 h. Liver tissue (300 mg) was homogenized with a tight dounce homogenizer in the presence of 0.1 M Tris-HCl buffer containing protease inhibitor cocktail followed by centrifugation at 100×g at 4°C for 5 min twice. The supernatants were incubated with biotinylated anti-GFP antibody (20 μg/1 ml) for 20 m at 4°C using a rotary shaker and the lysosomes-eGFP-Lamp1 complex was isolated with a streptavidin-magnetic bead (Pierce). The isolated lysosomal fractions were subjected to western blotting or metal analysis.

For in vitro metal transport assays, Hepa1c1c7 cells were transfected with different amounts of pCMV-Flag-Slc46a3 plasmid using JetPEI transfection reagent for 24 h. The cells were then treated with copper sulfate ($CuSO_4 \cdot 5H_2O$, 10 μg/ml) in serum-free DMEM media for 1 h. After extra incubation of cells in fresh serum-free DMEM for 30 m, the cells were lysed with M-PER buffer, and subjected to sequential fractionation by differential centrifugal at 800 g for 5 m, 13,000×g for 10 min, or 543,000×g for 30 min at 4°C. Each pellet or supernatant were subjected to western blotting and metal analysis.

**Reporter gene assays.** To verify the role of AhR on *Slc46a3* regulation, the putative DRE site (−17 kb) of *Slc46a3* was inserted in pGL4.10 with additional minimal TATA box sequence using AscI/PacI sites. pGL4.10-3×DREmini was depicted as wild-type and pGL4.10-3×mutDREmini was indicated as mutant as an experimental control. The inserted sequences for pGL4.10-3×DREmini and pGL4.10-3×mutDREmini are GGAG ATAGCGTGCCATGGTCTGCCTGGAGATAGCGTGCCATGGTCTGTTTGGAGA TAGCGTGCCATGGTCTGAGACACTAGAGGGTATATAATGGAAG CTCGACTT CCAGCT and TGGAGATAAAAAACCATGGTCTGCCTGGAGATAAAAAACCAT GGTCTGTTTGGAGATAAAAAACCATGGTCTGAGACACTAGAGGGTATATAA TGGAAGCTC GACTTCCAGCT, respectively. Hepa1c1c7 cells were transfected with the plasmids overnight using JetPEI reagent and treated with 10 nM TCDD for 6 h in the presence or absence of the AhR inhibitor CH-223191. The dual-luciferase assay was conducted using the Promega dual luciferase system. Renilla luciferase activity was used as transfection control.

**Histology.** For microscopic examination, fresh livers were fixed in 10% buffered formalin and embedded with paraffin. Tissue sections (4 μm) were stained with hematoxylin and eosin (HE) (Sigma-Aldrich, St. Louis, MO, USA). Frozen liver tissues were cut at 10 μm thickness and stained with Oil Red O to detect lipid droplets.

**Transmission electron microscopy (TEM)**. TEM images of livers were serviced by the Center for Cancer Research (CCR) of NCI at Frederick.

*Southern blot analysis*: Tail DNA from the WT and *Slc46a3*$^{-/-}$ mice was performed as described previously[38] using specific probes (Supplementary Table 1).

*Northern blot analysis*: Total RNA from different tissue samples was prepared by using Trizol reagent (Invitrogen, Carlsbad, CA). Northern blot analyses were performed as previously described[31]. *Slc46a3* cDNA (901 bp) was used as a Northern probe.

**ATP measurement**. Hepa1c1c7 cells were transfected with pEGFP (8 μg) or pEGFP-Slc46a3 (8 μg) plasmids in 100 mm dishes overnight and cell lysates and ATP was measured according to the manufacturer's instructions (DoGenBio, Seoul, Korea).

**Statistical analysis**. Experimental values are expressed as mean ± SD. Statistical analysis was performed by one-way ANOVA for multiple comparisons using Tukey's multiple comparisons test, two-way ANOVA for comparisons of multi factors using Bonferroni's method, or two-tailed Student's *t* test for unpaired data. $P < 0.05$ was considered statistically significant. GraphPad PRISM v.8.4.3 was used for statistical analysis.

## Data availability

All relevant data are available from the corresponding authors upon request. Source data are provided with this paper. Supplementary information is provided in Supplementary Figs. 1–9, Supplementary Table 1. The mouse liver transcription profiling by array for Fig. 1c was downloaded from Gene Expression Omnibus under the accession code GSE55084 and AhR ChIP-seq data in Fig. 1d was downloaded from Gene Expression Omnibus under the accession code GSE97634. Source data are provided with this paper.

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

## Acknowledgements

This work was supported by the National Research Foundation of Korea (NRF) grant funded by the Korea government (MEST) (NRF-2015R1A5A2008833), the Basic Science Research Program through the National Research Foundation of Korea (NRF) funded by the Ministry of Science, ICT & Future Planning (2015R1C1A1A02036595), National Institutes of Health Grant DK079209 (to J.L.), National Institutes of Health Grant GM79465 (to C.J.C.), and the National Cancer Institute Intramural Research Program.

## Author contributions

J.H.K. performed animal experiments and cell experiments as well as molecular biology. T.M. analyzed the *Slc46a3*$^{-/-}$ phenotype and lipid using the liver and serum samples. C.

F.F. and K.H. helped with the computational analysis. J.L. helped with metal analysis. H.Y. analyzed TG. T.N. followed the generation of *Slc46a3*$^{-/-}$ mice and evaluated the phenotype of *Slc46a3*$^{-/-}$ mice. S.Y. discovered that *Slc46a3* was an AhR target gene and generated the *Slc46a3*$^{-/-}$ mice. S.J. synthesized the copper sensing probe. J.H.K., D.K., and K.W.K. performed metabolomics analysis. S.J. and C.J.C. contributed to the copper detection experiments. J.H.K. and F.J.G. wrote the manuscript and supervised the study with contributions from all authors.

## Competing interests

The authors declare no competing interests.
