## [Peer Review File · Nature Communications]

Reviewers' Comments:

Reviewer #1:

Remarks to the Author:

Following the observation that 2,3,7,8-tetrachlorodibenzo-p-dioxin (TCDD) elicited liver-specific induction of Slc46a3 mRNA, the authors explored potential roles in triglyceride accumulation using knock-out and overexpression models that suggested a putative role in lysosomal Cu transport. Studies revealed co-localization of eGFP-SLC46A3 and the lysosomal membrane protein LAMP1 in mouse Hepa1c1c7 cells and characterized changes in Cu levels using genetic, ectopic expression, probes, and chelators in mouse liver and hepatocyte models that were further supported by molecular dynamic simulations. Alterations to mitochondrial morphology associated with Slc46a3 manipulation were also reported. To test whether SLC46A3 was involved in TCDD-elicited steatosis, loss of Slc46a3 was shown to reduced triglyceride accumulation on a high fat diet model suggesting a role in lipid homeostasis, possibly through disruption of energy metabolism mediated by AMPK. The manuscript suggests the previously uncharacterized solute transporter, SLC46A3, may play a role in energy homeostasis.

However, conclusions regarding AhR-mediated regulation of Slc46a3 and effects of TCDD on Cu transport/homeostasis need to be strengthened. Evidence supporting the induction of Slc46a3 by TCDD is modest relative to Cyp1a1/1a2. Further information regarding the dose level and dosing regimen used in the reported studies is needed. Reporting the induction level of Slc46a3 in other published hepatic gene expression datasets that used TCDD or other AHR agonists could provide additional supportive data. This should include in vitro and in vivo dose response studies and time course studies as well as studies that used diverse AhR agonists (PCDFs, PCBs, PAHs) and relevant controls (PCB153). Datasets in mouse, rat and human models can be found in the Gene Expression Omnibus, TGGates db and DrugMatrix db. In addition, the authors should also consider the effects of TCDD on other genes/proteins associated with Cu transport/homeostasis (i.e., Slc3a1/2, Css, Cp, Atp7a/b) as well as other other metals including Fe and Zn that could affect the transport of Cu. Finally, there is a lack of experimental detail, especially regarding the TCDD studies, to critically evaluate the data in order to assess the conclusions regarding AHR regulation of Slc46a3.

Other Comments:

- There is some confusion in the Introduction on how Slc46a3 was identified as a gene of interest. Lines 89 – 95 suggest that a cDNA microarray analysis was performed on Ahr-null and Arnt liver-specific null mice identified 1200006F02Rik, yet no microarray data is presented. Is this published data? What is the source? This data should be included in the Results section.
- Materials and methods: Two different study designs are described for TCDD treatment (24h at 10ug/kg or 7 days at 200 ng/kg). It is not clear whether the 7 days treatment group was given a single dose, injected daily, or other. Furthermore, none of the data for TCDD treatment indicates under which treatment condition they were determined (e.g. Fig 6a,b). A diagram depicting experimental design would be helpful.
- Materials and methods: The authors should consider depositing the plasmids used in the presented study and/or their sequence to a plasmid repository such as Addgene to promote reproducibility.
- Statistical Analysis: Authors indicate only the use of one-way ANOVA and t-test, but some analyses would have required a two-way ANOVA (e.g. TCDD treatment of genetic models where the two factors are genetic background and treatment).
- The authors focus on hepatic effects in Slc46a3^{-/-} mice on the basis that it is induced only in this tissue by TCDD. However, online databases such as biogps would suggest that other tissues express Slc46a3 to an equal or greater extent in mice (e.g. small intestine and dendritic cells). While the use of Hepa1c1c7 cells certainly points to hepatic function, perturbation of lipid accumulation is only observed in vivo where knock-out and overexpression are not liver specific. Can the authors comment on the putative role of other tissues, evidence for or against this, and why a whole-body knockout model was used rather than a hepatocyte-specific model such as the ArntLiv model?
- Is a ~30% increase in hepatic Cu levels physiologically significant. How does this compare to

other diseases or disease models. For example, Cu levels in Wilson's and Menke disease.

- The writing and organization of the manuscript is confusing. Major editorial review is needed for the Introduction. Figure panels should be presented in the same order as discussed. The first figure cited in the manuscript is Figure 1h, and it is not until a following section that Figure 1a is presented.
- L280-282: The ChIP-seq data used for dioxin response element site identification only shows the location of protein binding and does not identify the DRE DNA sequence. The authors should clarify how DREs were identified and if AhR binding is concurrent with DRE locations.
- Cu accumulation increased ~30% in perfused livers of Slc46a3^{-/-} mice. The authors suggest this "might be due to Cu accumulation in lysosomes". However, Cu levels were also significantly increased in lysosomes isolated from the liver expressed Slc46a3 (with the use of plasmid Dsred-SLC46A3; Fig. 2b). Does absence or presence of Slc46a3 increase Cu level?
- Fig. S2 shows higher Fe levels in Slc46a3^{-/-} knockout mice compared to Cu levels. Was there an effect on Zn? Cu homeostasis is linked to Fe and Zn homeostasis. Dysregulation in one may dysregulate the others and should be discussed.
- Was the increased in Slc26a3 mRNA expression after TCDD treatment also observed in Hepa1c1c7 cells?
- What software was used for SLC46A3 model construction? How was the accuracy of model was verified?

Minor Comments:

- Materials and methods: Indicate catalog number of antibodies to ensure reproducibility of results.
- L303: "Cu levels in the pellet at 13,000 g (Figure 1h)..." should be 2c.
- Figure 1: Figure 1d (liver) and Figure 1e (left plot) appear to be redundant. Please comment on why it appears that the same endpoint was assessed twice?
- Figure 2c and S6: There are random asterisks. What do these mean?
- Figure 3i: The figure legend indicates a significant increase, yet no error bars or evident of statistical test is shown. Was only 1 replicate used for this evaluation?
- Fig. 4: Cu ions are hard to see in the protein models. Please indicate location with arrow, or increasing the size of the Cu ions.
- Figures 6a,f, 7b: Hepatic triglyceride levels are shown using two different units. Please correct or justify the use of different units.
- There are no references to supplementary figures 2, 4, 5 and 6 in the text.
- In the figures 6 and S1 authors used abbreviation "KO" which probably refers to Slc46a3^{-/-} mice. The abbreviation occurs only in these 2 places. Suggest replacing "KO" with Slc46a3^{-/-} to be consistent.
- Figure 3: there is no "f" letter in figure description.
- Figure S7: Abbreviation "TEPT" should be defined in the figure description.
- Figure S11: There is missing "A" letter in the figure
- Some identified typographical errors include:
 - o L219: dunce should be dounce
 - o L280: Cyp1a1 should be italicized
 - o L244: the most conserved residues are positioned towards inside the fold
 - o L481-482: accumulate in liver by reduce mitochondrial function
 - o The aryl hydrocarbon receptor is abbreviated as AhR (e.g. L70), and AHR (e.g. L82).

Reviewer #2:

Remarks to the Author:

This study provides an impressive set of the data, which reveal SLC46A3 to operate as a new copper (Cu) transporter. Using a wide range of methods, the authors show that SLC46A3 resides in the lysosomes, where it promotes Cu translocation from the cytosol to the lysosomal lumen. These conclusions were further supported by observations (i) that SLC46A3 knockout leads to

accumulation of Cu in hepatocytes and (ii) that elevated SLC46A3 expression reduces bioavailable Cu, which is needed to support mitochondrial function. Altogether, the main findings of the paper uncover a novel Cu-transporting function of SLC46A3 and provide new important insights in our understanding of Cu metabolism in health and disease. Therefore, the manuscript might be of great interest to the readership of the Nature Communications. However, in my view, in its current form the manuscript needs revision to address the below comments.

- 1) It turns out that both ATP7B and SLC46A3 sequester Cu to lysosomes. However, SLC46A3 apparently does not compensate for the loss of ATP7B function in Wilson disease. I think that the authors have to explain why?
- 2) Page 15. Lysosome size increase in SLC46A3-overexpressing cells. Any idea why?
- 3) Figure 5C is confusing. The authors look at red mitotracker in the cells expressing red version of SLC46A3. I would remove this figure because the subsequent figure 5D basically shows the right GFP version of SLC46A3 in the same experiment with similar outcome.
- 4) Along the same line, I am not sure that mitotracker is the best marker to judge about the functional state of mitochondria. I suggest to use TMRE-mitochondrial membrane potential dye.
- 5) The quality of EM images in the Figs. 5G, H is very low. I can recognize neither mitochondrial cristae nor the membrane (both external and internal). These images have to be improved for publication.
- 6) Existing literature suggests that mitochondria suffer from both lack and excess of copper. The authors show that overexpression of SLC46A3 causes dysfunction of mitochondria, while it is unclear whether this is the case for SLC46A3 suppression. Is the intracellular accumulation of copper in SLC46A3-deficient cells sufficient to cause mitochondrial dysfunction?
- 7) Does the copper excess in SLC46A3-deficient cells activate autophagy? Recent studies suggest that copper activates autophagy through several mechanisms (Polishchuk et al., 2019; Tsang et al., 2020). Thus, one might expect reduced autophagy in the case of SLC46A3 overexpression and an increase in autophagy in the case of SLC46A3 suppression.

Reviewer #3:

Remarks to the Author:

In this study Kim et al. hypothesize that SLC46A3-mediated solute transport may alter hepatic lipid homeostasis and play a role in the progression of NAFLD. The authors identify copper as a possible substrate for lysosomal SLC46A3 and use computational modeling to simulate potential copper binding sites on the transporter. Using SLC46A3 KO mice, Kim et al. show that SLC46A3 deletion alters mitochondrial morphology and reduces hepatic lipid accumulation, protecting SLC46A3 KO mice from diet-induced hepatic steatosis. In contrast, overexpression of the transporter impaired oxygen consumption and induced hepatic lipid accumulation.

Comments

Overall, this is an interesting and carefully performed study that addresses an interesting question. The authors provide strong evidence to suggest that lysosomal SLC46A3-mediated copper transport plays a role in regulating hepatic mitochondrial function and hepatic lipid metabolism. I have a few suggestions that would strengthen this paper.

1. Are there any baseline differences in body composition, energy intake, energy expenditure

between WT and SLC46A3^{-/-} mice prior to high fat feeding?

2. As mentioned in the Discussion, copper is shown to play a role in the regulation of lipolysis. Does SLC46A3 expression regulate intrahepatic lipolysis? And if so is hepatic ATGL phosphorylation altered in SLC46A3 KO mice?

3. It would be of interest to know if SLC46A3 expression impacts other bioactive lipid species (e.g. diacylglycerols, ceramides) as well as hepatic insulin sensitivity.

4. The authors show that cytochrome c oxidase protein levels are unchanged in the SLC46A3 KO (Figure 3). Does overexpression of SLC46A3 alter protein levels of cytochrome c oxidase?

5. The authors adequately describe the effects of copper deficiency on liver metabolism, but the mechanism by which SLC46A3 deletion and increased intracellular copper increases mitochondrial activity is unclear.

Reviewer #4:

Remarks to the Author:

The authors deorphanize the solute carrier transporter SLC46A3 and characterize its pathophysiological role, by using a range of genetic, cellular, biochemical, and computational methods. The results indicate that SLC46A3 is located in the liver serving as regulator of lysosomal copper transport that controls triglyceride levels. Overall the study addresses an important problem in a comprehensive fashion, and reveals a novel role for SLC46A3 in disease. Not too much is known about the function of many human SLCs and their role in disease, so this study is timely. However, I have major concerns regarding the structural modeling and its interpretation, as well as the written English of the manuscript.

1) The authors built a structural model of SLC46A3 based on a homolog structure of a plant protein, and use this model to support SLC46A3's interaction with copper (via docking). However, SLC46A3 shares sequence identity of only 10% with the template structure, far below what is considered the "twilight zone" for homology modeling. This suggests that the model is unlikely to be sufficiently accurate for the analysis performed in this study. Critically, docking calculations are almost guaranteed to be incorrect for this sort of model, as described in numerous studies in the field. This voids any conclusions made from Figure 4.

2) While it is plausible that the overall architecture or the fold of the model is correct for this similarity level, additional information should be provided about the model to convince me that this is indeed the case. We need to know: what the sequence identity is within the binding site? What is the score of the model? Did the authors use any additional assessment criteria? This overall presentation of the model is very lacking.

3) It should also be noted that MD simulations on a low quality model cannot yield any conclusive results. Multiple studies have shown that MD simulations do not improve model accuracy for low-quality homology models. In fact, even for high quality models or experimentally determined structures, simulations can negatively affect docking results. Therefore, The conclusions based on Figure 5 are not supported by the presented data.

4) The manuscripts requires significant editing. Specifically, the language style requires clarification/editing (possibly by a third party) to improve the quality and flow of the manuscript. Below highlighted a few examples. While each example is minor, taken together, this manuscript requires a major rewrite.

Minor Comments:

1. Line 42 pg 3 "SLC46A" should be "SLC46A3".

2. Lines 47-48 pg3. Change "forced expression" to overexpression and it should be mitochondrial membrane.

3. Line 67. Consider changing to "Nonalcoholic fatty liver disease (NAFLD) is one of the most

common chronic liver diseases in developed countries." It strengthens your point.

4. Lines 68-69—The authors add additional causes of fatty liver which is a repetitive point. The paragraph already starts with causes of fatty liver. Please combine with line 64..

5. Line 69. Pay attention to subject-verb agreement.

6. Line 69. It is unclear which "activations" the authors refer to.

7. Line 70- AHR is introduced. "aryl hydrocarbon receptor (AhR)" and in line 81-82 "aryl-hydrocarbon receptor (AHR)" it is given again with a different acronym and naming – be consistent with hyphens and acronyms and only provide whole names and corresponding acronyms at the first use.

8. Lines 87-89. Improve clarity.

9. Line 94- Consider reordering the panels in Figure 1 so the reader starts with panels a-c rather than d-f. In general, reorder all Fig. 1 panels to reflect the flow of the text.

10. Lines 95-100. Improve clarity.

11. Lines 323-324. This should be SLC46A3 in both instances.

12. Line 338 SLC6A3 should be SLC46A3

Reviewers' Comments

Reviewer #1 (Remarks to the Author):

Following the observation that 2,3,7,8-tetrachlorodibenzo-p-dioxin (TCDD) elicited liver-specific induction of *Slc46a3* mRNA, the authors explored potential roles in triglyceride accumulation using knock-out and overexpression models that suggested a putative role in lysosomal Cu transport. Studies revealed co-localization of eGFP-SLC46A3 and the lysosomal membrane protein LAMP1 in mouse Hepa1c1c7 cells and characterized changes in Cu levels using genetic, ectopic expression, probes, and chelators in mouse liver and hepatocyte models that were further supported by molecular dynamic simulations. Alterations to mitochondrial morphology associated with *Slc46a3* manipulation were also reported. To test whether SLC46A3 was involved in TCDD-elicited steatosis, loss of *Slc46a3* was shown to reduced triglyceride accumulation on a high fat diet model suggesting a role in lipid homeostasis, possibly through disruption of energy metabolism mediated by AMPK. The manuscript suggests the previously uncharacterized solute transporter, SLC46A3, may play a role in energy homeostasis.

However, conclusions regarding AhR-mediated regulation of *Slc46a3* and effects of TCDD on Cu transport/homeostasis need to be strengthened.

1) Evidence supporting the induction of *Slc46a3* by TCDD is modest relative to *Cyp1a1/1a2*.

Answer: According to analysis of the ChIP-seq data (GEO, GSE97634), it's possible that *Slc46a3* can be regulated by AhR through an enhancer in part because the AhR DNA-binding sequence was found at -17 from the start site and ,135 kb upstream from the start codon. *Cyp1a1* and *Cyp1a2* are a bona-fide AhR target genes, as revealed by numerous studies of over the past 35 years. To confirm that is an AhR target gene, we carried out AhR-luciferase reporter gene assays in Hepa1c1c7 cells after treatment with TCDD. AhR-luciferase activity was significantly increased by TCDD and the activity was abolished by mutation of the AhR-binding site AATGGAGATAgcgtgCCATGGTCTG to AATGGAGATAaaaaaCCATGGTCTG. The TCDD-induced *Slc46a3* mRNA level was also decreased by the AhR inhibitor (CH-223191) in Hepa1c1c7 cells. New data were added to Figures 1f, 1g, and 1h

2) Further information regarding the dose level and dosing regimen used in the reported studies is needed.

Answer: We analyze the effect of TCDD on *Slc46a3* mRNA expression at different times and doses. *Slc46a3* mRNA was increased by TCDD (10 ug/kg) time-dependently within 24 h of dosing. However, there were no differences between the time regimens (day 3 and day 7) and doses (10 or 200 ug/kg) for induction of *Slc46a3* mRNA. All the similar levels of *Slc46a3* mRNA by TCDD were maintained 24 h after TCDD treatment. This result is lin line for the high affinity of TCDD affinity for AhR and that target gene activation persists for many days and weeks after exposure. New data were added to Figures 1j and 1k.

3) Reporting the induction level of *Slc46a3* in other published hepatic gene expression datasets that used TCDD or other AHR agonists could provide additional supportive data. This should include in vitro and in vivo dose response studies and time course studies as well as studies that

used diverse AhR agonists (PCDFs, PCBs, PAHs) and relevant controls (PCB153). Datasets in mouse, rat and human models can be found in the Gene Expression Omnibus, TGGates db and DrugMatrix db.

Answer: We monitored the expression levels of *Slc46a*, *Cyp1a2* and *Ahr* mRNAs at different developing ages (1-8 weeks). Basal induction of both *Slc46a3* and *Cyp1a1*, but not *Ahr* mRNAs, were gradually increased up to 4 weeks in liver. This might indicate that *Slc46a3* could function at different developing stages. In addition, we extracted some data from the Gene Expression Omnibus datasets from GSE97634 (ChIP-seq) and GSE127217 (RNA-seq). Based upon the analysis of these datasets, *Slc46a3* is induced by TCDD. In addition, as the reviewer suggests, we tried to extract information using reported data sets on the AhR agonist (PCBFs, PCBs, PAHs) to analyze *Slc46a3* induction in different species including human and rat. However, we failed to find the *Slc46a3* gene name in the list of sets (for human hepatocytes, GEO, GSE14553 and for Rat hepatocytes, GEO: GSE14554). Although we failed to extract of some data from human and rat data about on *Slc46a3*, PCB-153, as a negative control of TCDD, was evaluated for the expression of *Slc46a3* in liver. According to the GEO data set (Accession: GSE55084), we confirmed that PCB-153 (80 mg/kg) does not induce the *Slc46a3* mRNA in liver when compared to TCDD (10 ug/kg). The effect of PCB-153 on *Slc46a3* mRNA expression was equivalent to the oil control. The data were added to Figure 1d. Furthermore, we used the AhR inhibitor CH-223191 to support the relationship between *Slc46a3* induction and AhR. As noted previously, CH-22191 inhibited TCDD-derived AhR-luciferase activity and *Slc46a3* induction. The data are shown in Figure 1f, 1g and 1h.

4) In addition, the authors should also consider the effects of TCDD on other genes/proteins associated with Cu transport/homeostasis (i.e., *Slc3a1/2*, *Css*, *Cp*, *Atp7a/b*) as well as other metals including Fe and Zn that could affect the transport of Cu.

Answer: We measured the levels of copper-related mRNAs. The data was added to Figure 4f.

5) Finally, there is a lack of experimental detail, especially regarding the TCDD studies, to critically evaluate the data in order to assess the conclusions regarding AHR regulation of *Slc46a3*.

Answer: AhR(DRE)-luciferase reporter gene assays were conducted in Hepa1c1c7 cells. A putative 3xAhRE (DRE) sequence and its mutant were applied and an AhR inhibitor was used to support the view that *Slc46a3* is an AhR target gene. Other factors also could play a role in *Slc46a3* induction because basal *Slc46a3* is relatively highly induced only in liver and small intestine. While we largely focused on the functional role of SLC46A3 in copper and lipid accumulation, further studies are warranted to identify transcriptional factors that control tissue-specific expression of *Slc46a3*. Additional supporting data was added to Figures 1f, 1g, and 1h

Other Comments:

- There is some confusion in the Introduction on how *Slc46a3* was identified as a gene of interest. Lines 89 – 95 suggest that a cDNA microarray analysis was performed on *Ahr*-null and *Arnt* liver-specific null mice identified 1200006F02Rik, yet no microarray data is presented. Is this published data? What is the source? This data should be included in the Results section.

Answer: Previously, we are executed old sets of microarray before re-naming *Slc46a3* from 1200006F02Rik. As an update of the gene list, currently we used the gene name as *Slc46a3*. We deleted 1200006F02Rik in the Introduction. RNA-seq data is already published (GEO,

GSE127217) and expression levels of *Slc46a3* after TCDD administration is similar to our microarray using the liver RNA from mice treated with TCDD (provided upon request)

- Materials and methods: Two different study designs are described for TCDD treatment (24h at 10ug/kg or 7 days at 200 ng/kg). It is not clear whether the 7 days treatment group was given a single dose, injected daily, or other.

Answer: Corrected.

Furthermore, none of the data for TCDD treatment indicates under which treatment condition they were determined (e.g. Fig 6a,b). A diagram depicting experimental design would be helpful.

Answer: This was added to Figure 6.

- Materials and methods: The authors should consider depositing the plasmids used in the presented study and/or their sequence to a plasmid repository such as Addgene to promote reproducibility.

Answer: We will deposit the plasmids we used in this study.

- Statistical Analysis: Authors indicate only the use of one-way ANOVA and t-test, but some analyses would have required a two-way ANOVA (e.g. TCDD treatment of genetic models where the two factors are genetic background and treatment).

Answer: Some figures such as Figures 1c and 6a were analyzed with two-way ANOVA.

- The authors focus on hepatic effects in *Slc46a3*^{-/-} mice on the basis that it is induced only in this tissue by TCDD. However, online databases such as biogps would suggest that other tissues express *Slc46a3* to an equal or greater extent in mice (e.g. small intestine and dendritic cells).

Answer: According to our preliminary experiments, northern blotting results show that basal levels of *Slc46a3* mRNA differs in various tissues (Figure below). Relatively high *Slc46a3* mRNA was measured in both liver and small intestine. However, only TCDD induced *Slc46a3* mRNA in liver. As BioGPS, some cells such as dendritic cells and glands such as salivary and lacrimal glands express high level of *Slc46a3* mRNA. This information will be useful for other specific studies.

While the use of Hepa1c1c7 cells certainly points to hepatic function, perturbation of lipid accumulation is only observed *in vivo* where knock-out and overexpression are not liver specific. Can the authors comment on the putative role of other tissues, evidence for or against this, and why a whole-body knockout model was used rather than a hepatocyte-specific model such as the ArntLiv model?

Answer: In this experiment, we are focused the function of SLC46A3 in liver because *Slc46a3* is induced by TCDD only in the liver. However, determined the whole phenotypical change in mice using the whole-body knock-out system. Among other tissues, liver had the largest phenotypical change. Of course, abnormal induction from the normal condition or ablation of gene function by gene disruption would result in different features in different tissues or cells. Although we did not provide the results using a liver-specific *Slc46a3* knock-out model, we believe that the whole body knockout is sufficient to offer clues to any novel SLC46A3 function. Induction of *Slc46a3* mRNA by TCDD in the Hepa1c1c7 cells (≤ 2 fold induction) or primary hepatocyte is not similar to that of liver (≥ 4 fold induction). The reason for this discrepancy between *in vitro* and *in vivo* is not known. However, the overexpression system using cells may provide more similar results as found in *in vivo* to determine SLC46A3 function. Prolonged exposure of copper may change hepatic or intestinal physiology in the absence of SLC46A3. This will be subject of future studies

- Is a ~30% increase in hepatic Cu levels physiologically significant. How does this compare to other diseases or disease models. For example, Cu levels in Wilson's and Menke disease.

Answer:

Answer: As subtle change of intracellular copper levels is correlated with SLC46A3 induction or loss, this may induce a pathological effect similar to that of Menkes disease or Nieman-Pick type C. Changes of copper levels may induce changes in cell membranes or lysosomal lipid contents such as sphingomyelins and ceramides. Thus, we previously monitored sphingomyelins using the isolated lysosomal fractions after SLC46A3 overexpression. Interestingly, lysosomal sphingomyelin was changed by forced SLC46A3 overexpression. Thus, we are investigating this in future studies.

- The writing and organization of the manuscript is confusing. Major editorial review is needed for the Introduction. Figure panels should be presented in the same order as discussed. The first figure cited in the manuscript is Figure 1h, and it is not until a following section that Figure 1a is presented.

Answer: Corrected.

- L280-282: The ChIP-seq data used for dioxin response element site identification only shows the location of protein binding and does not identify the DRE DNA sequence. The authors should clarify how DREs were identified and if AhR binding is concurrent with DRE locations.

Answer: The putative AhR binding sequences were addressed in the Figure. The role of putative AhR binding sequence was verified using AhR-luciferase reporter gene assays in the Hepa1c1c7 cells treated with TCDD. The data are displayed in Figures 1d, 1f, and 1g.

- Cu accumulation increased ~30% in perfused livers of *Slc46a3*^{-/-} mice. The authors suggest this "might be due to Cu accumulation in lysosomes". However, Cu levels were also significantly increased in lysosomes isolated from the liver expressed *Slc46a3* (with the use of plasmid Dsred-SLC46A3; Fig. 2b). Does absence or presence of *Slc46a3* increase Cu level?

Answer: Corrected to “might be due to Cu accumulation in the cytosol.”

- Fig. S2 shows higher Fe levels in *Slc46a3* -/- knockout mice compared to Cu levels. Was there an effect on Zn? Cu homeostasis is linked to Fe and Zn homeostasis. Dysregulation in one may dysregulate the others and should be discussed.

Answer: It's possible that copper levels could affect to iron or zinc homeostasis in liver or other tissues. According to the ICP data, only iron was significant increased by disruption of *Slc46a3*. In additional experiments, ferritin light polypeptide 1 (*Ftl1*) mRNA was significantly increased in mice liver after TCDD (10 ug/kg) treatment for 7 days. The results suggest that decreased copper could induce the iron-related transporting system. Additional data was added to Figure 4f.

- Was the increased in *Slc26a3* mRNA expression after TCDD treatment also observed in Hepa1c1c7 cells?

Answer: We measured *Slc26a3* mRNA in the Hepa1c1c7 cells and liver after TCDD treatment. However, the *Slc26a3* mRNA was not changed by TCDD; even endogenous *Slc26a3* mRNAs was barely detected (data is not shown). Thus, chloride transporting system is not changed by TCDD.

- What software was used for SLC46A3 model construction? How was the accuracy of model was verified?

Answer: For computational modeling of protein structure and docking simulation, we removed the data in accordance with the editor's suggestion, because it does not necessary lead to determining the physiological role of SLC46A3.

Minor Comments:

- Materials and methods: Indicate catalog number of antibodies to ensure reproducibility of results.

Answer: All catalog numbers are indicated in the Reporting summary file.

- L303: “Cu levels in the pellet at 13,000 g (Figure 1h)...” should be 2c.

Answer: Corrected

- Figure 1: Figure 1d (liver) and Figure 1e (left plot) appear to be redundant. Please comment on why it appears that the same endpoint was assessed twice?

Answer: The figure has been deleted.

- Figure 2c and S6: There are random asterisks. What do these mean?

Answer: Asterisks in bands indicate the FLAG-SLC46A3 protein shown in labelling.

- Figure 3i: The figure legend indicates a significant increase, yet no error bars or evident of statistical test is shown. Was only 1 replicate used for this evaluation?

Answer: The wording in the text was changed to “CCS protein level was increased after treatment with BCS~”. This figure was re-arranged based on additional data.

- Fig. 4: Cu ions are hard to see in the protein models. Please indicate location with arrow, or increasing the size of the Cu ions.

Answer: The computer modeling data was removed as suggested by the editor.

- Figures 6a,f, 7b: Hepatic triglyceride levels are shown using two different units. Please correct or justify the use of different units.

Answer: Corrected.

- There are no references to supplementary figures 2, 4, 5 and 6 in the text.

Answer: Corrected and figure numbers rearranged due to additional data.

- In the figures 6 and S1 authors used abbreviation “KO” which probably refers to Slc46a3^{-/-} mice. The abbreviation occurs only in these 2 places. Suggest replacing “KO” with Slc46a3^{-/-} to be consistent.

Answer: Corrected

- Figure 3: there is no “f” letter in figure description.

Answer: Corrected

- Figure S7: Abbreviation “TEPT” should be defined in the figure description.

Answer: Corrected

- Figure S11: There is missing “A” letter in the figure

Answer: Corrected

Some identified typographical errors include:

- L219: dunce should be dounce

Answer: Corrected

- L280: Cyp1a1 should be italicized

Answer: Corrected

- L244: the most conserved residues are positioned towards inside the fold

Answer: Deleted

- L481-482: accumulate in liver by reduce mitochondrial function

Answer: Corrected

- The aryl hydrocarbon receptor is abbreviated as AhR (e.g. L70), and AHR (e.g. L82).

Answer: Corrected

Reviewer #2 (Remarks to the Author):

This study provides an impressive set of the data, which reveal SLC46A3 to operate as a new copper (Cu) transporter. Using a wide range of methods, the authors show that SLC46A3 resides in the lysosomes, where it promotes Cu translocation from the cytosol to the lysosomal lumen. These conclusions were further supported by observations (i) that SLC46A3 knockout leads to accumulation of Cu in hepatocytes and (ii) that elevated SLC46A3 expression reduces bioavailable Cu, which is needed to support mitochondrial function. Altogether, the main findings of the paper uncover a novel Cu-transporting function of SLC46A3 and provide new important insights in our understanding of Cu metabolism in health and disease. Therefore, the manuscript might be of great interest to the readership of the Nature Communications. However, in my view, in its current form the manuscript needs revision to address the below comments.

1) It turns out that both ATP7B and SLC46A3 sequester Cu to lysosomes. However, SLC46A3 apparently does not compensate for the loss of ATP7B function in Wilson disease. I think that the authors have to explain why?

Answer: Here, we measured mRNA levels for some copper-related transporters such as *Ctr1*, *Atp7a*, and *Atp7b* as well as other transporters for such those for iron and zinc. The results were added to the text. For the copper transporter, *Atp7b* mRNA, it was not changed by TCDD (10ug/kg) in liver. For Wilson disease, the loss of ATP7b function results in accumulation of copper in the liver. We believe that elimination of excess copper may not be enough by SLC46A3. Interestingly, we extracted data on the level of *Slc46a3* mRNA in a previous study using a mouse model (*Atp7b*^{-/-}). The GEO data set (GSE125637) shows that *Slc46a3* mRNA was slightly increased with significance (p<0.05, n=4). This suggests that excess copper also affects the level of SLC46A3 to eliminate the copper but elimination rate might not be enough.

2) Page 15. Lysosome size increase in SLC46A3-overexpressing cells. Any idea why?

Answer: For enlarged lysosome by SLC46A3, we propose that it might be due to the abnormal accumulation of copper leading to hypertrophy. This might also affect to the increase of lysosomal protein such as LAMP1. Thus, we measured the LAMP1 protein in the TCDD-treated mouse liver. We found that glycosylated LAMP1 was significantly increased. This result was added to Figure S7 and described in the text. However, future studies will be done to explain the phenomenon.

3) Figure 5C is confusing. The authors look at red mitotracker in the cells expressing red version of SLC46A3. I would remove this figure because the subsequent figure 5D basically shows the right GFP version of SLC46A3 in the same experiment with similar outcome.

Answer: Empirically, Mitotracker signal overrides the signal of mCherry-SLC46A3. Thus, we measured the expression of mCherry-Slc46a3 or mCherry in the primary hepatocyte after hydrodynamic injection. and incubated with Mitotracker.

4) Along the same line, I am not sure that mitotracker is the best marker to judge about the functional state of mitochondria. I suggest to use TMRE-mitochondrial membrane potential dye.

Answer: TMRE is a potential dye for the measurement of mitochondrial potential. Thus, we used TMRE in Figure 5e. Interestingly, we have used the mitotracker to exam morphology of mitochondria. However, the Mitotracker signal also changed by eGFP-SLC46A3 induction. In the text, "Here Mitotracker can be used as a mitochondrial potential marker." was deleted.

5) The quality of EM images in the Figs. 5G, H is very low. I can recognize neither mitochondrial cristae nor the membrane (both external and internal). These images have to be improved for publication.

Answer: The figures were replaced with high resolution figures.

6) Existing literature suggests that mitochondria suffer from both lack and excess of copper. The authors show that overexpression of SLC46A3 causes dysfunction of mitochondria, while it is unclear whether this is the case for SLC46A3 suppression. Is the intracellular accumulation of copper in SLC46A3-deficient cells sufficient to cause mitochondrial dysfunction?

Answer: Copper as well as other metal ions is critical for maintaining the function of mitochondria depending on the levels. Here, we believe that SLC46A3 deficiency may increase the mitochondrial function positively by increasing the mitochondrial potential and results in increased energy metabolism as shown in the present data. It seems that copper is not accumulated to pathophysiological levels as in Wilson disease. However, the chronic state of this condition could change to any phenotypic with increased age. Thus, we monitor the phenotypic change at different levels in different ages.

7) Does the copper excess in SLC46A3-deficient cells activate autophagy? Recent studies suggest that copper activates autophagy through several mechanisms (Polishchuk et al., 2019; Tsang et al., 2020). Thus, one might expect reduced autophagy in the case of SLC46A3 overexpression and an increase in autophagy in the case of SLC46A3 suppression.

Answer: We measured LC3-II in the presence or absence of SLC46a3 using liver and tissue culture cells. However, no significant differences were measured. It might be due to the levels of copper in the cell. The western blot data for LC3-II measurement was added to Figure 4e.

Reviewer #3 (Remarks to the Author):

In this study Kim et al. hypothesize that SLC46A3-mediated solute transport may alter hepatic lipid homeostasis and play a role in the progression of NAFLD. The authors identify copper as a possible substrate for lysosomal SLC46A3 and use computational modeling to simulate potential copper binding sites on the transporter. Using SLC46A3 KO mice, Kim et al. show that SLC46A3 deletion alters mitochondrial morphology and reduces hepatic lipid accumulation, protecting SLC46A3 KO mice from diet-induced hepatic steatosis. In contrast, overexpression of the transporter impaired oxygen consumption and induced hepatic lipid accumulation.

Comments:

Overall, this is an interesting and carefully performed study that addresses an interesting question. The authors provide strong evidence to suggest that lysosomal SLC46A3-mediated copper transport plays a role in regulating hepatic mitochondrial function and hepatic lipid metabolism. I have a few suggestions that would strengthen this paper.

1. Are there any baseline differences in body composition, energy intake, energy expenditure between WT and SLC46A3^{-/-} mice prior to high fat feeding?

Answer: We carried out glucose tolerance test (GTT) in WT and *Slc46a3*^{-/-} mice under a normal chow diet (8-week-old male mice). Also, we monitored the level of glucose levels in blood after oral administration of glucose (1 g/kg). However, there were no significant changes between the two groups. The results were added to Figure S3.

2. As mentioned in the Discussion, copper is shown to play a role in the regulation of lipolysis. Does SLC46A3 expression regulate intrahepatic lipolysis? And if so is hepatic ATGL phosphorylation altered in SLC46A3 KO mice?

Answer: Here, we postulate that SLC46A3 may not induce intrahepatic lipolysis. We examined the level of p-ATGL in both WT and *Slc46a3*^{-/-} mice. However, western bands were barely detected and no significant changes were found.

These data were not added to the manuscript

3. It would be of interest to know if SLC46A3 expression impacts other bioactive lipid species (e.g. diacylglycerols, ceramides) as well as hepatic insulin sensitivity.

Answer: In a preliminary study, we found that sphingomyelin levels in lysosomes and cytosol were changed in the SLC46a3-overexpressing cells. For hepatic insulin sensitivity, we tested GTT, however, no significant differences were found between the groups.

4. The authors show that cytochrome c oxidase protein levels are unchanged in the SLC46A3 KO (Figure 3). Does overexpression of SLC46A3 alter protein levels of cytochrome c oxidase?

Answer: Cytochrome c oxidase (COX IV) levels were measured in the eGFP-SL46A3-expressing Hepalcl cells, however, no significant difference were found. This data was included in Figure 4k.

5. The authors adequately describe the effects of copper deficiency on liver metabolism, but the mechanism by which SLC46A3 deletion and increased intracellular copper increases mitochondrial activity is unclear.

Answer: We present some mitochondrial study for the measurement of mitochondrial oxygen consumption rates (OCR) and mitochondrial membrane potential using TMRE, It's possible to predict the dysfunction of mitochondria by SLC46A3 overexpression. As the reviewer suggests, we measured the relative level of ATP to determine whether ATP was reduced by overexpression of eGFP-SLC46A3 in the Hepal1c7 cells. ATP levels were significantly reduced by eGFP-SLC46A3. SLC46A3-derived copper deficiency affects the function of mitochondria. The new data were added to Figure 5g.

Reviewer #4 (Remarks to the Author):

The authors deorphanize the solute carrier transporter SLC46A3 and characterize its pathophysiological role, by using a range of genetic, cellular, biochemical, and computational methods. The results indicate that SLC46A3 is located in the liver serving as regulator of lysosomal copper transport that controls triglyceride levels. Overall the study addresses an important problem in a comprehensive fashion, and reveals a novel role for SLC46A3 in disease. Not too much is known about the function of many human SLCs and their role in disease, so this study is timely. However, I have major concerns regarding the structural modeling and its interpretation, as well as the written English of the manuscript.

1) The authors built a structural model of SLC46A3 based on a homolog structure of a plant protein, and use this model to support SLC46A3's interaction with copper (via docking). However, SLC46A3 shares sequence identity of only 10% with the template structure, far below what is considered the "twilight zone" for homology modeling. This suggests that the model is unlikely to be sufficiently accurate for the analysis performed in this study. Critically, docking calculations are almost guaranteed to be incorrect for this sort of model, as described in numerous studies in the field. This voids any conclusions made from Figure 4.

Answer: Answer: Modeling data was deleted as suggested by the editor.

2) While it is plausible that the overall architecture or the fold of the model is correct for this similarity level, additional information should be provided about the model to convince me that this is indeed the case. We need to know: what the sequence identity is within the binding site? What is the score of the model? Did the authors use any additional assessment criteria? This overall presentation of the model is very lacking.

Answer: Modeling data was deleted as suggested by the editor..

3) It should also be noted that MD simulations on a low quality model cannot yield any conclusive results. Multiple studies have shown that MD simulations do not improve model accuracy for low-quality homology models. In fact, even for high quality models or experimentally determined structures, simulations can negatively affect docking results. Therefore, The conclusions based on Figure 5 are not supported by the presented data.

Answer: Modeling data was deleted as suggested by the editor..

4) The manuscripts requires significant editing. Specifically, the language style requires clarification/editing (possibly by a third party) to improve the quality and flow of the manuscript.

Below highlighted a few examples. While each example is minor, taken together, this manuscript requires a major rewrite.

Answer: The writing was revised by a professional editor.

Minor Comments:

1. Line 42 pg 3 “SLC46A” should be “SLC46A3”.

Answer: Corrected.

2. Lines 47-48 pg3. Change “forced expression” to overexpression and it should be mitochondrial membrane.

Answer: Corrected.

3. Line 67. Consider changing to “Nonalcoholic fatty liver disease (NAFLD) is one of the most common chronic liver diseases in developed countries.” It strengthens your point.

Answer: Corrected.

4. Lines 68-69—The authors add additional causes of fatty liver which is a repetitive point. The paragraph already starts with causes of fatty liver. Please combine with line 64.

Answer: Corrected.

5. Line 69. Pay attention to subject-verb agreement.

Answer: Corrected.

6. Line 69. It is unclear which “activations” the authors refer to.

Answer: Corrected. Activations was changed to effects.

7. Line70- AHR is introduced. “aryl hydrocarbon receptor (AhR)” and in line 81-82 “aryl-hydrocarbon receptor (AHR)” it is given again with a different acronym and naming – be consistent with hyphens and acronyms and only provide whole names and corresponding acronyms at the first use.

Answer: Corrected.

8. Lines 87-89. Improve clarity.

Answer: Changed to “As part of NAFLD, it is necessary to understand the mechanism of toxicity of TCDD how it induces fatty liver accumulation as result of induction of AhR-responsive genes”

9. Line 94- Consider reordering the panels in Figure 1 so the reader starts with panels a-c rather than d-f. In general, reorder all Fig. 1 panels to reflect the flow of the text.

Answer: Corrected. Figure 1 was re-organized.

10. Lines 95-100. Improve clarity.

Answer: Changed

11. Lines 323-324. This should be SLC46A3 in both instances.

Answer: Corrected.

12. Line 338 SLC6A3 should be SLC46A3
Answer: Corrected.

Reviewers' Comments:

Reviewer #1:

Remarks to the Author:

The authors have sufficiently addressed my concerns in the revised manuscript.

Reviewer #2:

Remarks to the Author:

The authors addressed all my comments and added several important experiments that improved the overall quality of the manuscript. I have only very minor comment.

Page 14. The authors say "...copper may affect hepatic autophagy" but do not provide any reference that in my view should be included.

Reviewer #3:

Remarks to the Author:

The authors have satisfactorily addressed all of my comments.

Reviewers' Comments

REVIEWERS' COMMENTS

Reviewer #1 (Remarks to the Author):

The authors have sufficiently addressed my concerns in the revised manuscript.

Reviewer #2 (Remarks to the Author):

The authors addressed all my comments and added several important experiments that improved the overall quality of the manuscript. I have only very minor comment.

Page 14. The authors say "...copper may affect hepatic autophagy" but do not provide any reference that in my view should be included.

Answer: reference was added in the text.

Reviewer #3 (Remarks to the Author):

The authors have satisfactorily addressed all of my comments.